# Comparisons of the tropospheric specific humidity from GPS radio occultations with ERA–Interim, NASA MERRA and AIRS data

Panagiotis Vergados[1], Anthony J. Mannucci[1], Chi O. Ao[1], Olga Verkhoglyadova[1], and Byron Iijima[1]

[1] Jet Propulsion Laboratory, California Institute of Technology, Pasadena, California, USA

**Corresponding author:** P. Vergados, Jet Propulsion Laboratory, M/S 138-310B, 4800 Oak Grove Dr., Pasadena, CA, 91109, USA. (Panagiotis.Vergados@jpl.nasa.gov)

**Abstract.** We construct a 9–year data record (2007-2015) of the tropospheric specific humidity (SH) using Global Positioning System radio occultation (GPS RO) observations from the Constellation Observing System for Meteorology, Ionosphere, and Climate (COSMIC) mission. This record covers the ±40$^o$ latitude belt and includes estimates of the zonally averaged monthly mean SH from 700 hPa up to 400 hPa. It includes three major climate zones: a) the deep tropics (±15$^o$), b) the trade winds belts (±15–30$^o$), and c) the subtropics (±30–40$^o$). Our objective is to compare the RO observations with the European Center for Medium-range Weather Forecasts Re-Analysis Interim (ERA-Interim), the Modern-Era Retrospective analysis for Research and Applications (MERRA), and the Atmospheric Infrared Sounder (AIRS) to examine the consistency among the data sets. We present RO SHs from both JPL and UCAR processing centers to provide an estimate of the structural uncertainty of the RO SH products. The results show that the RO observations capture the seasonal and interannual SH variability as all other data sets. On average, the JPL-RO SH agrees with both reanalyses to within 10%, is overall larger than all data sets, having maximum differences with AIRS by ~10–30%, and is almost twice as wet as all other data sets in the middle-to-upper troposphere at the subtropics. The UCAR-RO SH also agrees with both reanalyses and AIRS, but is systematically drier than all other data sets. Provided the estimated differences between the RO observations and the rest of the data sets, together with the retrieval uncertainty of the SH products from all data sets, we conclude that RO observations are a valuable independent observing system, which could augment independent reanalyses and satellite platforms. We anticipate that the COSMIC-2 mission will increase the observational sampling; thus, improving the coverage and quality of the observed SH climatology.

## 1 Introduction

The Intergovernmental Panel on Climate Change (IPCC) Fifth Assessment Report (AR5) [*Flato et al.*, 2013] documented that identifying the vertical structure of humidity is subject to great uncertainty, because dynamical processes that cannot be captured by one sensor alone drive water vapor. Hence, we ought to quantify and understand the degree of agreement of the water vapor concentration throughout the vertical extent of the troposphere among different sensors, in order to improve the representation of the Earth's atmospheric humidity content that is key to predicting future climate [*Hegerl et al.,* 2015].

To-date, ground- and space-based platforms, reanalyses, and model simulations do not provide precise knowledge of the water vapor's concentration, or its trends over time, in multiple regions of the Earth's atmosphere [*Sherwood et al.*, 2010]. This is because of the combination of different reasons that include: (a) sampling bias due to cloudiness, deep convection, or surface emissivity variations; (b) biases due to limited local time coverage, or random observations versus volume-filling scans; (c) coarse spatial resolutions, and (d) misrepresentation of the planetary boundary layer's (PBL) moisture content [*Hannay et al.*, 2009] that induces errors in the lower-to-middle troposphere moist convection.

In particular, infrared (IR) space-based platforms have a coarse vertical resolution (e.g., 2.0–3.0 km), are prone to cloud contamination [*Fetzer et al.,* 2006], and tend to be low biased over wet and dry humidity extremes [*Fetzer et al.,* 2008; *Chou et al.*, 2009]. The use of IR observations in the lower troposphere still remains a challenge, due to the decreasing information content and the difficulty detecting low-cloud contamination [*Schreier et al.,* 2014]. Space-based microwave (MW) sounders, despite having low sensitivity to precipitation and clouds, have a coarse vertical resolution (e.g., 3.0 km in case of the Microwave Limb Sounder (MLS) [*Waters*

*et al.,* 2006]) and are sensitive to the a–*priori* solution that could cause unsuccessful limb-viewing radiance retrievals (e.g., of up to 30% in the case of MLS [*Read et al.,* 2007]) under clear sky but moist conditions. Heavy cloudiness, especially in the middle-to-upper troposphere can also introduce biases in the upwelling MW radiation from water vapor due to the presence of ice particles that can contaminate the MW retrievals [*Fetzer et al.,* 2008]. Global Circulation Models (GCMs) do not properly represent the middle troposphere moist convection [*Sherwood et al.,* 2004; *Holloway and Neelin,* 2009; *Frenkel et al.,* 2012], and large discrepancies in the tropospheric humidity among different reanalyses [*Chen et al.,* 2008] and among reanalyses, models, and satellite observations [*Chuang et al.,* 2010; *Jiang et al.,* 2012; *Tian et al.,* 2013; *Wang and Su,* 2013] still persist.

The path towards constraining the models, reanalyses, and satellite water vapor observations uncertainty is to compare them against data sets that are as independent from their *a-priori* information as possible. Here, we exploit the multi-year record of Global Positioning System Radio Occultation (GPS RO) observations for remote sensing the Earth's water vapor content. GPS ROs offer unique atmospheric observing properties such as, all–weather sensing, high vertical resolution (100–200 m; *Kursinski et al.* [2000]; *Schmidt et al.* 2005]), high specific humidity (SH) accuracy (< 1.0 g/Kg), and full diurnal cycle sampling.

The description of the humidity retrieval process from RO observations is discussed in details in *Kursinski et al.* [1997], *Kursinski and Hajj* [2001], and *Collard and Healey* [2003], to name a few. Numerous authors have validated these products against reanalyses, satellite observations, and radiosondes as discussed in *Steiner et al.* [1999], *Gorbunov and Kornblueh* [2001], *Divakarla et al.* [2006], *Ho et al.* [2007], *Chou et al.* [2009], *Ho et al.,* [2010], *Sun et al.* [2010], *Gorbunov et al.* [2011], *Kishore et al.,* [2011], *Wang et al.* [2013], *Vergados et al.*

[2014], *Vergados et al.* [2015]. Also, recently, *Kursinski and Gebhardt* [2014] proposed a novel approach to further improve the retrieved humidity distribution from ROs in the middle troposphere. Motivated by the above studies, our primary objective is to create a short-term SH data record (9 years) based on RO observations and compare it against NASA's Modern Era Retrospective Analysis for Research and Applications (MERRA), European Center for Medium-range Weather Forecasts Reanalysis Interim (ERA–Interim), and Atmospheric Infrared Sounder (AIRS) data sets. Our goal is to evaluate the consistency of the RO SH with respect to state-of-the-art reanalyses and satellite observations by quantifying the RO SH differences with the rest of the data sets over the tropics and subtropics We anticipate to gain new insights about the SH distribution over different convective regions, which could provide guidelines for future model improvements. The uniqueness of this study is that it is the first to compare nearly a decade long data records of RO SH information and their interannual variability against MERRA, ERA–Interim, and AIRS. Of importance is the fact that we use MERRA, instead of MERRA-2, because MERRA does not assimilate ROs (unlike ERA–Interim), providing an independent data set when comparing the RO SH observations. Section 2 presents the data sets we use in this analysis together with their retrieval characteristics. In Section 3, we present and discuss the RO SH climatologies with respect to the rest of the data sets. Section 4 summarizes our current research.

## 2 Methodology

We create time series of tropospheric SH climatologies using the COSMIC observations (using both the UCAR and the JPL retrievals), the MERRA and ERA-Interim data sets, and the Atmospheric Infrared Sounder (AIRS) observations. These climatologies contain a 9-year

measurement record from January 2007 until December 2015 and represent monthly zonal mean averages. We study the tropics and subtropics (±40°, in three distinct latitudinal regions) from 700 hPa up to 400 hPa, because this region is key to climate research [*IPCC,* 2007], but models and observations have large SH differences in the middle and upper troposphere [e.g., *Jiang et al.,* 2012; *Tian et al.,* 2013; *Wang and Su,* 2013], and we select this pressure range because the RO SH retrievals are most robust.

### 2.1 Constellation Observing System for Meteorology, Ionosphere and Climate

The COSMIC constellation of six microsatellites were launched in April 2006 orbiting the Earth at an altitude of ~800 km in near-circular Low Earth Orbit (LEO) [*Anthes et al.,* 2008]. They measure the phase and amplitude of the transmitted dual frequency *L*-band GPS signals ($f_1$=1.57542 GHz; $f_2$=1.22760 GHz) as a function of time. The relative motion of the COSMIC satellites with respect to the GPS satellites and the presence of the atmosphere cause a Doppler frequency shift on the transmitted GPS signals upon receipt at the COSMIC satellites. The magnitude of the Doppler frequency shift is estimated as the time derivative of the recorded GPS signal phases, which together with precise knowledge of the position and velocity information of both the COSMIC and the GPS satellites allows for the estimation of the amount of bending of the transmitted GPS signals due to the presence of the atmosphere, from which one can infer the air refractive index [*Kursinski et al.,* 1997]. In the lower troposphere, the bending angle is retrieved using radioholographic methods (such as canonical transform or full spectrum inversion) that eliminate errors due to atmospheric multipath [e.g., *Ao et al.,* 2003]. The relative motion of the COSMIC and GPS satellite pair allows for the vertical scanning of the atmosphere providing vertical profiles of atmospheric refractivity, which contain temperature and humidity

information.

We use GPS RO-derived SH products from both the UCAR and the JPL processing centers, which follow different processing techniques to retrieve the SH products. Although this study does not focus on these differences, we ought to note that UCAR adopts a variational assimilation method, which requires *a-priori* knowledge of the atmospheric water vapor content (provided by ERA-Interim), implying that the derived SH products may be subject to error characteristics of the initial humidity conditions. On the other hand, JPL uses the refractivity equation (along with the hydrostatic equation and equation of state) to estimate the water vapor pressure given *a-priori* knowledge of the air temperature [*Hajj et al.*, 2002]:

$$N = 77.6\frac{P}{T} + 3.73 \cdot 10^5 \frac{e}{T^2} \Leftrightarrow e = \frac{1}{3.73 \cdot 10^5}(NT^2 - 77.6PT) \quad [1]$$

Where *N* (unitless) is the refractivity, *P* (mbar) is the pressure, *T* (K) is the temperature, and *e* (mbar) is the GPS-RO-derived water vapor pressure. The retrieval errors of the JPL SH products do not contain *a-priori* humidity information, but are subject to errors in the *a-priori* temperature information, which is provided by the ECMWF Tropical Ocean and Global Atmosphere (TOGA) database. Because Eq. (1) requires that both the RO and the ECMWF data sets be reported at the same pressure levels, we interpolate the temperature profiles into the vertical grid of the RO profiles using linear interpolation. Currently, the JPL-retrieved COSMIC air refractivity profiles are provided at 200 m vertical resolution in the lower to middle troposphere.

## 2.2 Modern-Era Retrospective Analysis for Research and Application

We use the MERRA (v5.2.0) analysis that employs a 3-D variational assimilation technique based on the Gridpoint Statistical Interpolation (GIS) scheme with a 6-hour update cycle [e.g., *Wu et al.,* 2002]. It does not assimilate RO observations, and therefore, it is an independent dataset from COSMIC. We analyze the monthly gridded SH products given in a 1/2-degree x 2/3-degree latitude–longitude grid and 42 vertical pressure levels. In the troposphere, the vertical pressure resolution from the surface up to 700 hPa is 25 hPa, whereas from 700 hPa until 300 hPa the vertical resolution is 50 hPa. MERRA is a NASA analysis that assimilates satellite observations using the Goddard's Earth Observing System (GOES) version 5.2.0 Data Assimilation System (DAS) [*Rienecker et al.,* 2008]. Primarily, it assimilates radiances from AIRS, the Advanced Television and Infrared Observatory Spacecraft Operational Vertical Sounder (ATOVS), and the Special Sensor Microwave Imager (SSM/I), and figure 4 in *Rienecker et al.* [2011] provides a detailed list of the rest of the data sets that are assimilated.

## 2.3. European Center for Medium-Range Weather Forecasts Re-Analysis Interim

We use the ERA-Interim [*Dee et al.,* 2011], which uses a 4-D variational assimilation technique [*Simmons et al.*, 2005] to analyze a variety of observational data sets to predict the state of the atmosphere with accuracy similar to what is theoretically possible based on the error characteristics of the assimilated data [*Simmons and Hollingsworth*, 2002]. We analyze the monthly gridded SH products given in a 0.75 degree x 0.75 degree latitude-longitude grid and 20 pressure levels from 1000 hPa up to 300 hPa. The vertical resolution from the surface up to 750 hPa is 25 hPa, but the vertical resolution decreases to 50 hPa between 750 hPa and 300 hPa. The primary data sets assimilated in ERA-Interim are radiosonde humidity observations, AIRS and microwave radiances, and as of 11/2006 GPS-RO bending angle profiles.

### 2.4. Atmospheric Infrared Sounder

We use the AIRS/AMSU v6 Level-3 data [*Tian et al.,* 2013a] and analyze the monthly gridded SH product given in a 1-degree x 1-degree latitude-longitude grid, which extend from the surface up to 100 hPa in 12 vertical pressure levels (~ 2.0 km vertical resolution). The latest AIRS v6 SH products are now available at standard pressure levels. The vertical resolution is between the surface up to 850 hPa is 75 hPa; between 700 hPa and 300 hPa the vertical resolution decreases to 100 hPa, and above the 300 hPa pressure level up to 100 hPa the vertical resolution is 50 hPa. The AIRS physical retrievals use an IR–microwave neural net solution [*Blackwell et al.*, 2008] as the first guess for temperature and water vapor profiles based on MIT's stochastic cloud-clearing and neural network solution described in *Khan et al.* [2014].

### 2.5. Data Sources

The GNSS-RO SH products are publicly available through JPL Global Environmental & Earth Science Information System (GENESIS) portal at ftp://genesis.jpl.nasa.gov/pub/genesis/glevels/cosmic?postproc, as well as accessible via the publicly available Atmospheric Grid Analysis and Extraction Profile (AGAPE) web interface at https://genesis.jpl.nasa.gov/agape/. The AIRS/AMSU v6 Level-3 SH products are described in detail in *Tian et al.* [2013], and for our analysis we use the AIRX3STM v006 data downloadable from multiple different online tools, including the Simple Subset Wizard (SSW) at https://disc.gsfc.nasa.gov/SSW/ and the Mirador search base at https://mirador.gsfc.nasa.gov. From the MERRA SH products we use are the MAIMNPANA v5.2.0 files, which we downloaded from the SSW. The ERA-Interim SH products are publicly available at http://apps.ecmwf.int/datasets/data/interim-full-moda/levtype=sfc/.

## 2.6. Establishing Data Set Accuracy

*Kursinski et al.* [1995] estimated that GPS-RO water vapor profiles have an accuracy of 10–20% below 7.0 km (~5.0% within the boundary layer), and *Kursinski and Hajj* [2001] estimated RO SH differences of ~0.1 g/kg compared to ECMWF. GPS-RO air refractivity accuracy of <1.0% at 2.0 km altitude [*Schreiner et al.,* 2007] reduces to ~0.2% above 5.0 km [*Kuo et al.,* 2005]. Given the air refractivity accuracy and a temperature error of ± 1.0 K, the JPL-RO SH is retrieved within ~0.2–0.4 g/kg accuracy at the tropics [*Vergados et al.,* 2014]. MERRA assimilates various observational data sets and the SH accuracy is a function of the accuracy of the assimilated products. In general, the MERRA SH retrievals are accurate to ~20% [*Rienecker et al.*, 2011]. AIRS estimated SH product accuracies are typically ~25% at p > 200 hPa [*Fetzer et al.,* 2008], and ERA-Interim SH products have an estimated accuracy of ~7–20% in the tropical lower-to-middle troposphere [*Dee et al.,* 2011].

## 3. Results and Discussion

We divide this section into three sub-sections that represent the three tropical climate environments we analyze, each of which exhibits different atmospheric dynamic properties. In each sub-section, we study the long-term SH in terms of its: a) annual and interannual variability and trend, and b) deviations with respect to our center's SH values (JPL–RO). The time series represent monthly zonal averages of the SH at individual pressure levels from the lower up to the middle troposphere: 700 hPa, 600 hPa, 500 hPa, and 400 hPa. We do not extend our analysis at higher altitudes due to the small contribution of water vapor on to the RO observations. We use the JPL-RO SH values as reference to quantify all statistics with respect to the rest of the data

sets, and the differences between the JPL and the UCAR time series serve as a guideline of an estimate of the SH structural uncertainty.

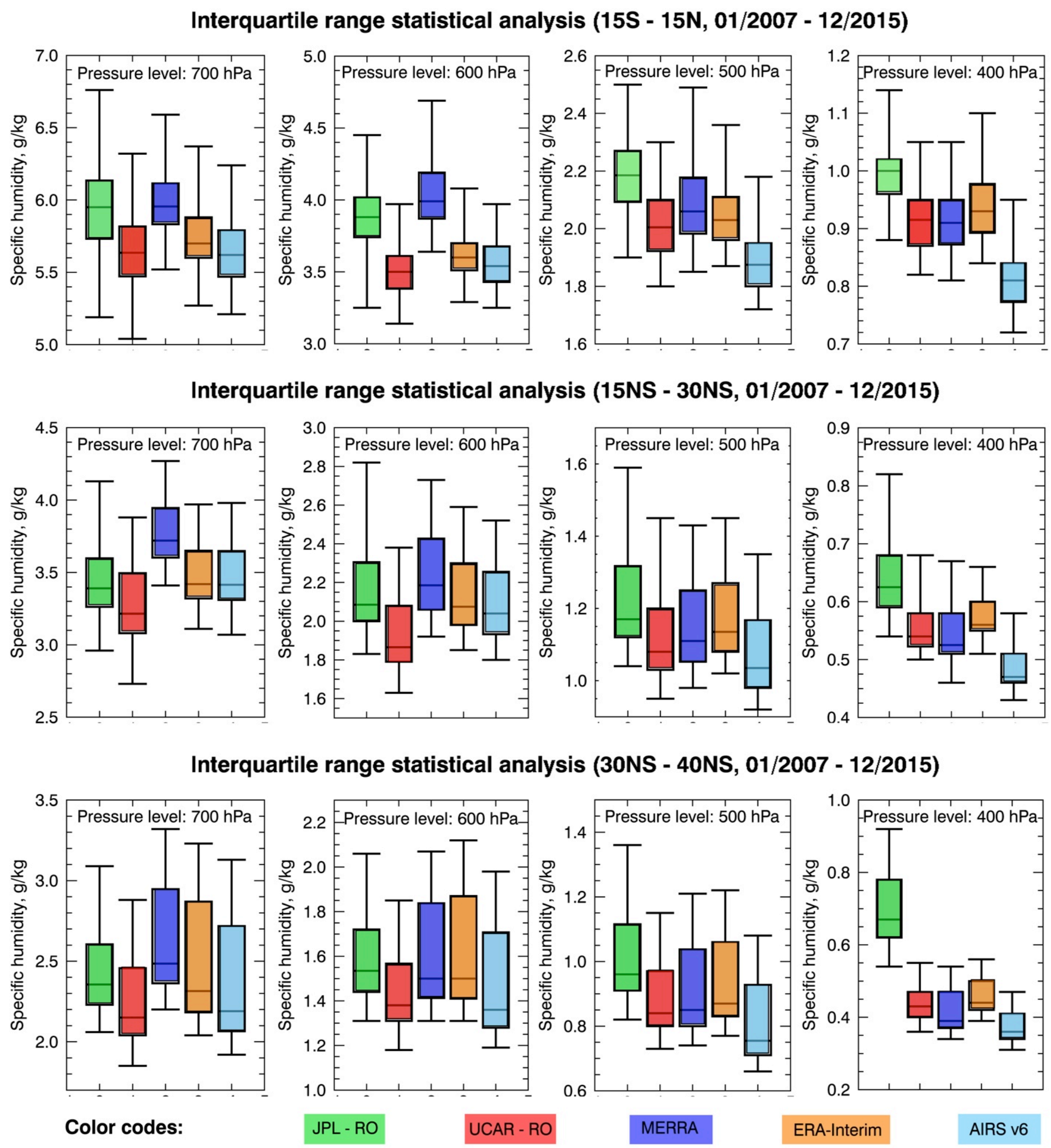

**Figure 1.** *Boxplots of the monthly zonal mean SH throughout the 2007–2015 time period for the 700 hPa, 600 hPa, 500 hPa, and 400 hPa over the ascending branch of Hadley cell (15S–15N) (top row), the trade winds belt (15NS–30NS) (middle), and the descending branch of Hadley cell at the subtropics (30NS–40NS) from JPL-RO (green), UCAR-RO (red), MERRA (blue), ERA–Interim (orange), and AIRS (cyan).*

### 3.1. Analysis of the SH at the ascending branch of the Hadley cell

The latitude belt within ±15$^{o}$ encompasses the ascending branch of the Hadley cell circulation. Moist air masses from both hemispheres converge within this narrow equatorial region, collide, and lead to heavy precipitation. The amount of the latent heat released during rainfall warms the air driving strong rising motions, deep convection, and high cloud formation.

The top row in figure 1 presents statistical information about the median, the interquartile range (IQR), and the minimum and maximum values of the SH time series over the entire observational record for all data sets throughout the vertical extent of the troposphere. Figure 2 shows details about the variability of the monthly zonal mean SH and Table 1 summarizes the results of figure 2.

**Table 1.** *Mean climatology, deviation of the mean climatology from JPL – RO, and linear regression fits of the SH time series from JPL–RO, UCAR–RO, ERA–Interim, MERRA, and AIRS over the 15S–15N climate region. The 2-sigma uncertainties are estimated for each statistical metric, and their statistical significance is evaluated at $p < 0.05$ confidence level. Boxes filled with red are statistically insignificant*

| **PART I:** | **9–year long mean of SH climatology with 2-sigma uncertainty, g/kg** | | | | |
|---|---|---|---|---|---|
| **Data Records** | **JPL–RO** | **UCAR–RO** | **ERA–Interim** | **MERRA** | **AIRS** |
| 400 hPa | 0.99 ± 0.12 | 0.92 ± 0.10 | 0.94 ± 0.12 | 0.91 ± 0.10 | 0.81 ± 0.08 |
| 500 hPa | 2.18 ± 0.26 | 2.01 ± 0.22 | 2.04 ± 0.22 | 2.08 ± 0.26 | 1.88 ± 0.20 |
| 600 hPa | 3.88 ± 0.44 | 3.51 ± 0.30 | 3.62 ± 0.30 | 4.03 ± 0.44 | 3.55 ± 0.32 |
| 700 hPa | 5.95 ± 0.60 | 5.64 ± 0.52 | 5.74 ± 0.46 | 5.99 ± 0.46 | 5.64 ± 0.44 |
| **PART II:** | **9–year long mean of deviations from JPL–RO, g/kg** | | | | |
| 400 hPa | n/a | - 0.08 | - 0.06 | - 0.08 | - 0.19 |
| 500 hPa | n/a | - 0.17 | - 0.14 | - 0.10 | - 0.31 |
| 600 hPa | n/a | - 0.37 | - 0.27 | + 0.15 | - 0.33 |
| 700 hPa | n/a | - 0.31 | - 0.22 | + 0.04 | - 0.32 |
| **PART III:** | **Linear regression fits of SH anomalies with 2-sigma uncertainty, g/kg/month** | | | | |
| 400 hPa | (1.0±3.0)x10$^{-4}$ | (3.7±2.2)x10$^{-4}$ | (2.4±2.2)x10$^{-4}$ | (0.1±2.1)x10$^{-4}$ | (0.3±2.0)x10$^{-4}$ |
| 500 hPa | (2.3±6.0)x10$^{-4}$ | (9.6±4.4)x10$^{-4}$ | (6.2±4.6)x10$^{-4}$ | (3.3±5.4)x10$^{-4}$ | (2.1±4.2)x10$^{-4}$ |
| 600 hPa | (-1.8±10)x10$^{-4}$ | (15.1±6.6)x10$^{-4}$ | (6.3±6.8)x10$^{-4}$ | (8.4±8.0)x10$^{-4}$ | (6.3±5.4)x10$^{-4}$ |
| 700 hPa | (6.1±12)x10$^{-4}$ | (17.2±9.0)x10$^{-4}$ | (14.1±8.8)x10$^{-4}$ | (1.3±7.2)x10$^{-4}$ | (12.9±7.2)x10$^{-4}$ |

In the lower troposphere, above the planetary boundary layer, the JPL-RO observations show almost the same mean SH value as MERRA ~ 6.0 g/kg (at 700 hPa) and ~ 4.0 g/kg (at 600 hPa) with the two data sets differing by < 1.0% and < 4.0% at the respective pressure levels (cf., Table 1) marking an excellent agreement between JPL–RO and MERRA. The UCAR–RO, AIRS, and ERA–Interim are in a very good agreement with one another differing by < 3.0% and all show that the lower troposphere is ~ 7.0–10% drier than what the JPL–RO and MERRA data sets indicate. This dryness is more pronounced at 600 hPa. These differences are statistically significant within the 2-sigma uncertainty. In the middle troposphere, at 500 hPa and 400 hPa, MERRA, ERA–Interim, and UCAR–RO agree very well capturing ~ 2.0–2.1 g/kg SH. However, the middle troposphere air appears to be moister in the JPL–RO data set than in the UCAR–RO and the two reanalyses by < 5.0–9.0%, which falls within the *Vergados et al.* [2014] uncertainty SH retrieval. AIRS is the driest among all data sets by < 10%, and its dryness becomes more apparent at 400 hPa. These discrepancies are statistically significant within the 2-sigma uncertainty.

The AIRS dry bias over the ITCZ [*Hearty et al.* 2014], possibly due to sampling limitations over cloud-covered regions, explains the observed systematic lower SH values with respect to all data sets over this deep convective environment. ERA–Interim underestimates the total cloud fraction over the ±15$^{o}$ region compared to MERRA [*Dolinar et al.,* 2016; figure 1] and is also colder than MERRA by ~1.0 K in the 2006–2011 time period at the tropics at 700 hPa [*Simmons et al.,* 2014; figure 18]. Given the definition of SH (as the product between the relative humidity and the saturation vapor pressure), it is evident why MERRA shows a wetter air than ERA–Interim in the lower troposphere.

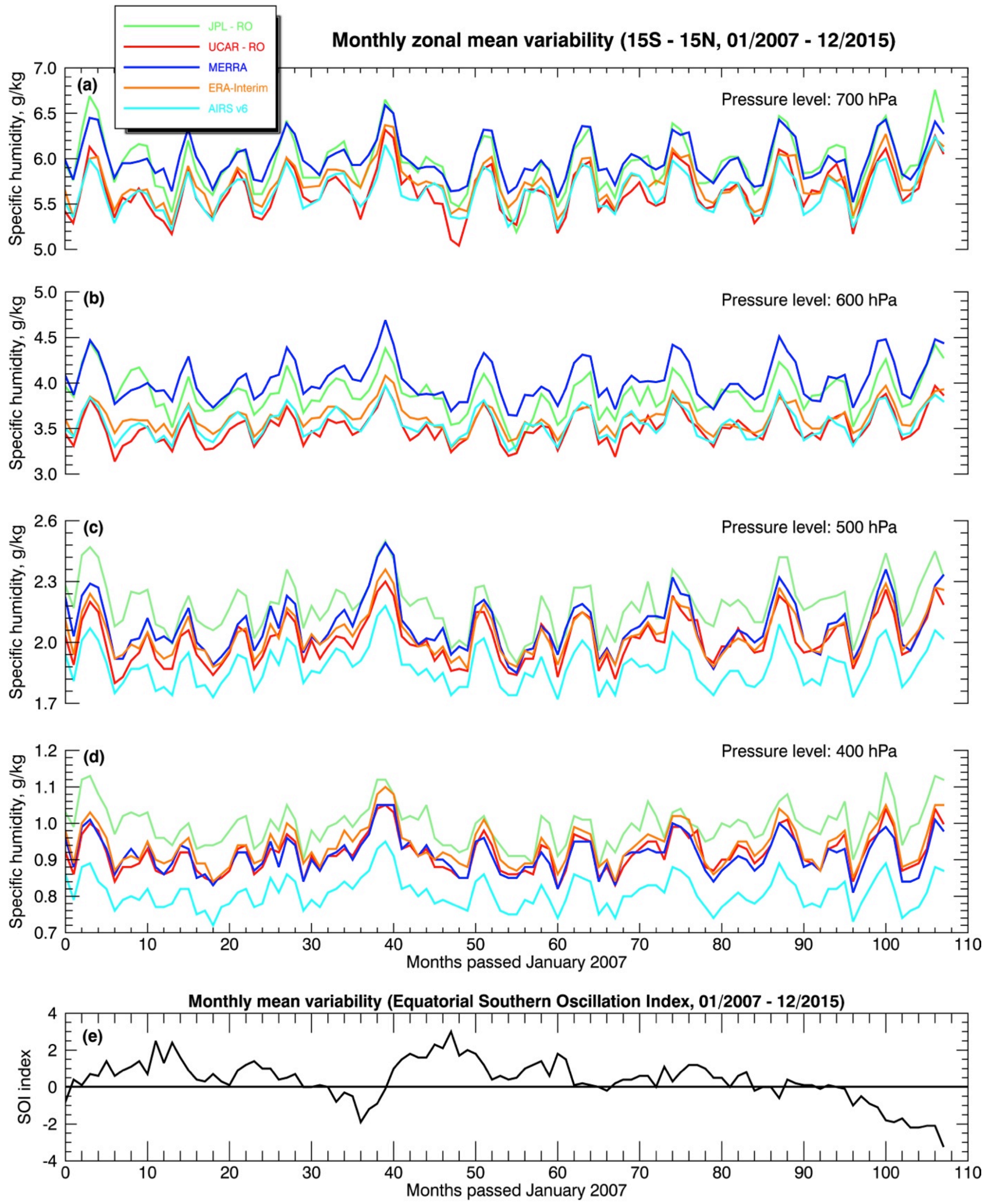

**Figure 2.** *Times series of the monthly zonal averages of the specific humidity from January 1, 2007 until December 31, 2015 from JPL–RO (green), UCAR – RO (red), ERA–Interim (orange), MERRA (blue) and AIRS (cyan) at (a) 500 hPa, (b) 400 hPa, (c) 700 hPa, and (d) 600 hPa pressure levels.*

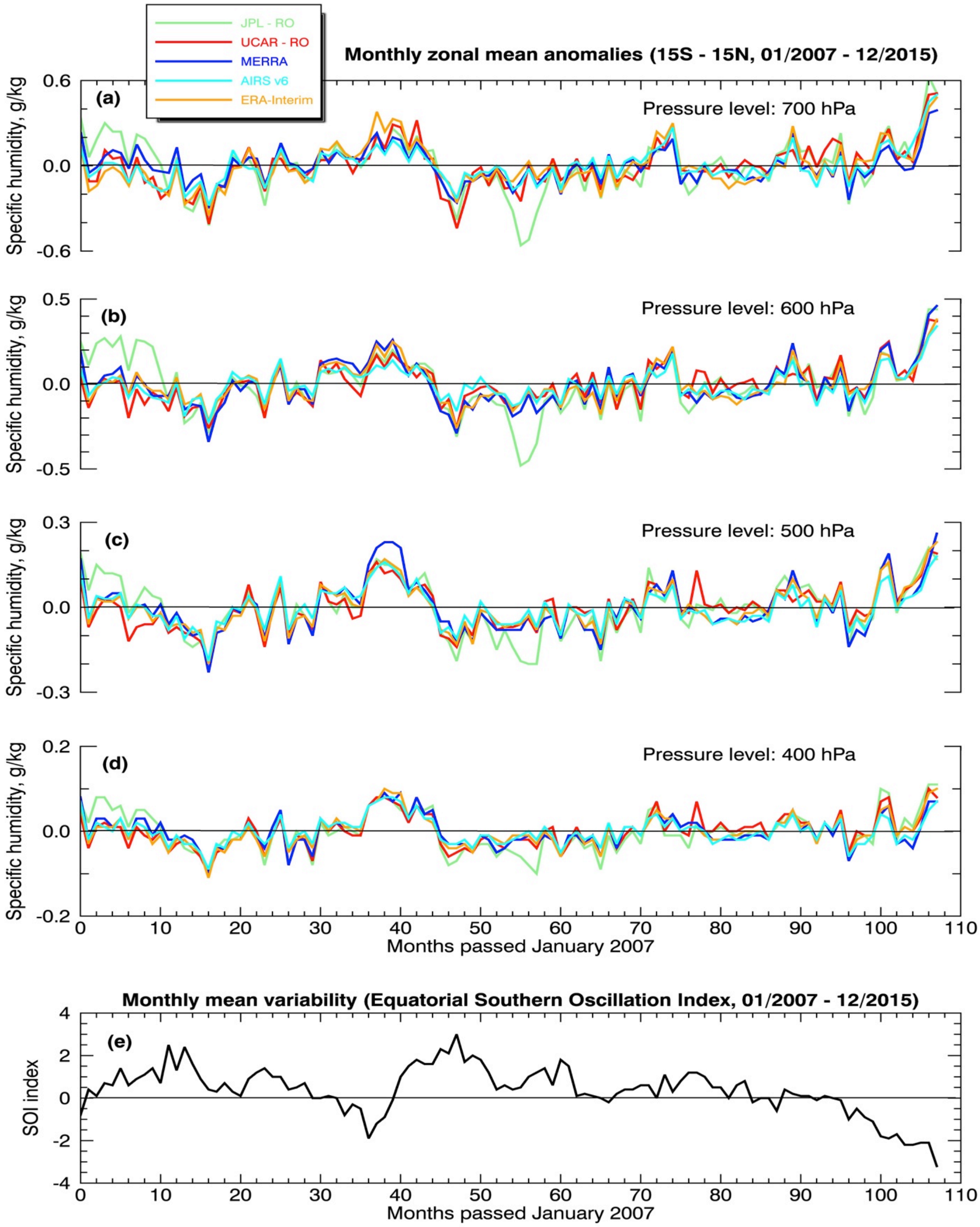

**Figure 3.** *Times series of the monthly zonal averages of the specific humidity interannual anomalies from January 1, 2007 until December 31, 2015 from JPL–RO (green), UCAR – RO (red), ERA–Interim (orange), MERRA (blue) and AIRS (cyan) at (a) 500 hPa, (b) 400 hPa, (c) 700 hPa, and (d) 600 hPa pressure levels.*

However, the cold bias in the ERA–Interim becomes small with altitude and reduces to almost zero at 500 hPa, and ERA–Interim starts showing a warm bias with respect to MERRA at 300 hPa by ~ 0.1–0.3 K [*Simmons et al.,* 2014]. This temperature bias between the two reanalyses could possibly explain why the two reanalyses begin to estimate similar SH values at 500 hPa and 400 hPa.

The fact that the UCAR–RO data set seems to consistently agree with ERA–Interim at all altitudes could be the result of the variational assimilation technique adopted by the UCAR center, which uses ERA–Interim humidity information as the *a-priori*. The systematic wetter air shown in the JPL SH values could be due to the warm bias in ERA–Interim above 500 hPa that leaks through the retrieval process of JPL's SH products (Eq. 1).

Despite the differences in the absolute value of the SH among the five different data sets, figure 2 shows that all data sets capture the same variability patterns, which exhibit clear signatures of an annual SH cycle. After computing the annual cycle for each data set and removing it from the time series, we estimate the respective SH interannual anomalies. The amplitude of these anomalies fluctuates around ±0.4 g/kg at 700 hPa, whose amplitude decreases to ±0.1 g/kg at 400 hPa. The interannual anomaly variations for all data sets in the middle troposphere correlate strongly (> 0.8) with those in the lower troposphere, but have smaller amplitude. The SH interannual anomalies for all data sets also show a moderate cross-correlation (> 0.5) with the monthly mean southern oscillation index (SOI), when using a 5–month lag, demonstrating that climate modes influence the troposphere in its entirety.

Based on a linear regression fit and a Student *t*-test statistical analysis at the 95% confidence level (criteria: $p < 0.05$ and 2-sigma) of the SH interannual anomalies, we find that JPL–RO and MERRA suggest no increase in the amount of SH with time between 700 hPa and

400 hPa (cf., Table 1). Contrary to that, the UCAR–RO and ERA–Interim data sets indicate a gradual increase of the absolute amount of SH throughout the vertical extend of the troposphere. The increase is faster at 700 hPa and slows down with height, with UCAR–RO systematically indicating faster moistening than ERA–Interim. The AIRS data sets show an increase of the SH at 700 hPa and 600 hPa at a rate similar to that of ERA–Interim, but no SH increase at 500 hPa and above.

### 3.2. Analysis of the SH at the trade winds zones

The ±15-30$^o$ belt, in both hemispheres, defines the trade winds zones, where dry air masses that descend from the Hadley cell at the subtropics travel towards the equator. These regions exhibit shallower convection compared to the ±15$^o$ region, as clouds forming in these regions are typically cumulus and do not extend above 4.0 km.

In the lower troposphere, above the boundary layer, we notice different behaviors in terms of the data sets' agreement compared to our analysis of the SH in the deep tropics. In particular, there is a statistically significant disagreement between the JPL–RO and MERRA data sets of ~ 10% (at 700 hPa) and ~ 3.5% (at 600 hPa), with MERRA being the wetter of the two. The JPL–RO data set agrees very well with both the ERA–Interim and the AIRS data sets having differences of ~ 1.0% (at 700 hPa) and ~ 2.0–3.0% (at 600 hPa); but, these difference are statistically insignificant. The UCAR–RO data set continues to be the driest among all data sets having statistically significant differences of ~ 15% (at 700 hPa and 600 hPa) and ~ 5.0% (at 700 hPa) to ~ 10% (at 600 hPa) with respect to MERRA and JPL–RO, respectively (cf., Table 2). In the middle troposphere, the summer season in the JPL–RO data set in noticeably wetter by ~ 4.0% than the rest of the data sets (cf., figure 4c) and this wetness becomes more pronounced at

400 hPa throughout the entire time period. Similar to the deep tropics, the UCAR–RO is in excellent agreement with both ERA–Interim and MERRA, and AIRS is still the driest data set and this dryness becomes more pronounced at 400 hPa.

**Table 2.** *Same as Table 1, but for the ±15-30° climate zone.*

| **PART I:** | **9–year long mean of SH climatology with 2-sigma uncertainty, g/kg** | | | | |
|---|---|---|---|---|---|
| **Data Records** | **JPL–RO** | **UCAR–RO** | **ERA–Interim** | **MERRA** | **AIRS** |
| 400 hPa | 0.64 ± 0.12 | 0.55 ± 0.08 | 0.57 ± 0.06 | 0.54 ± 0.10 | 0.49 ± 0.08 |
| 500 hPa | 1.22 ± 0.28 | 1.12 ± 0.24 | 1.17 ± 0.22 | 1.15 ± 0.24 | 1.07 ± 0.22 |
| 600 hPa | 2.17 ± 0.44 | 1.93 ± 0.38 | 2.13 ± 0.38 | 2.24 ± 0.42 | 2.09 ± 0.38 |
| 700 hPa | 3.44 ± 0.50 | 3.28 ± 0.54 | 3.48 ± 0.44 | 3.77 ± 0.44 | 3.48 ± 0.44 |
| **PART II:** | **9–year long mean of deviations from JPL–RO, g/kg** | | | | |
| 400 hPa | n/a | - 0.09 | - 0.07 | - 0.10 | - 0.16 |
| 500 hPa | n/a | - 0.11 | - 0.05 | - 0.07 | - 0.15 |
| 600 hPa | n/a | - 0.23 | - 0.02 | - 0.09 | - 0.07 |
| 700 hPa | n/a | - 0.16 | + 0.04 | + 0.33 | + 0.04 |
| **PART III:** | **Linear regression fits of SH anomalies with 2-sigma uncertainty, g/kg/month** | | | | |
| 400 hPa | $(-0.7\pm1.8)\times10^{-4}$ | $(1.1\pm1.2)\times10^{-4}$ | $(0.3\pm1.0)\times10^{-4}$ | $(-0.3\pm1.0)\times10^{-4}$ | $(-0.3\pm1.0)\times10^{-4}$ |
| 500 hPa | $(-0.5\pm3.6)\times10^{-4}$ | $(1.6\pm2.8)\times10^{-4}$ | $(-0.1\pm2.2)\times10^{-4}$ | $(-1.3\pm2.2)\times10^{-4}$ | $(-1.9\pm2.0)\times10^{-4}$ |
| 600 hPa | $(-6.9\pm6.6)\times10^{-4}$ | $(1.8\pm4.8)\times10^{-4}$ | $(-1.9\pm3.4)\times10^{-4}$ | $(-5.0\pm3.8)\times10^{-4}$ | $(-5.2\pm3.2)\times10^{-4}$ |
| 700 hPa | $(-3.9\pm8.6)\times10^{-4}$ | $(-0.4\pm7.2)\times10^{-4}$ | $(-3.8\pm4.8)\times10^{-4}$ | $(-7.5\pm4.6)\times10^{-4}$ | $(-6.2\pm4.4)\times10^{-4}$ |

Compared to the ±15° region, the absolute differences of the SH values averaged over the entire time period between the JPL–RO and the rest of the data sets throughout the vertical extent of the troposphere is smaller, except at the 400 hPa where it remains almost the same. Overall, this suggests that over less convective regions different data sets tend to agree better, signifying that convection is a limiting factor in properly sensing the amount of water vapor in the atmosphere. The monthly zonal mean SH variability also shows a clear annual cycle signature throughout the vertical extent of the troposphere, but the amplitudes of the SH interannual anomalies is ~ 50% smaller (cf., figure 5) than those estimated over the deep tropics.

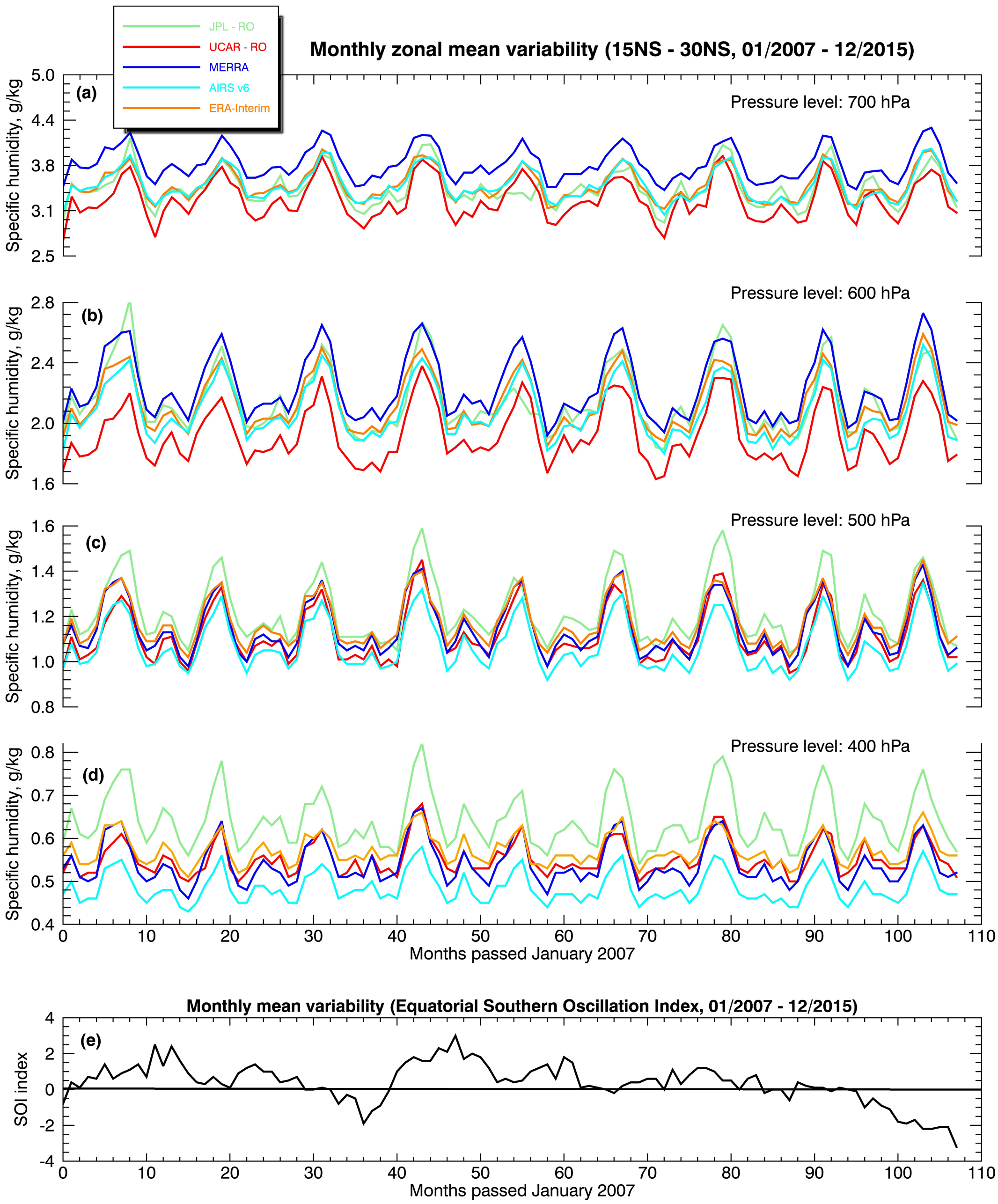

338

339 **Figure 4.** *Same as figure 2, but the results reflect SH trends in the 15NS–30NS latitudinal belt.*

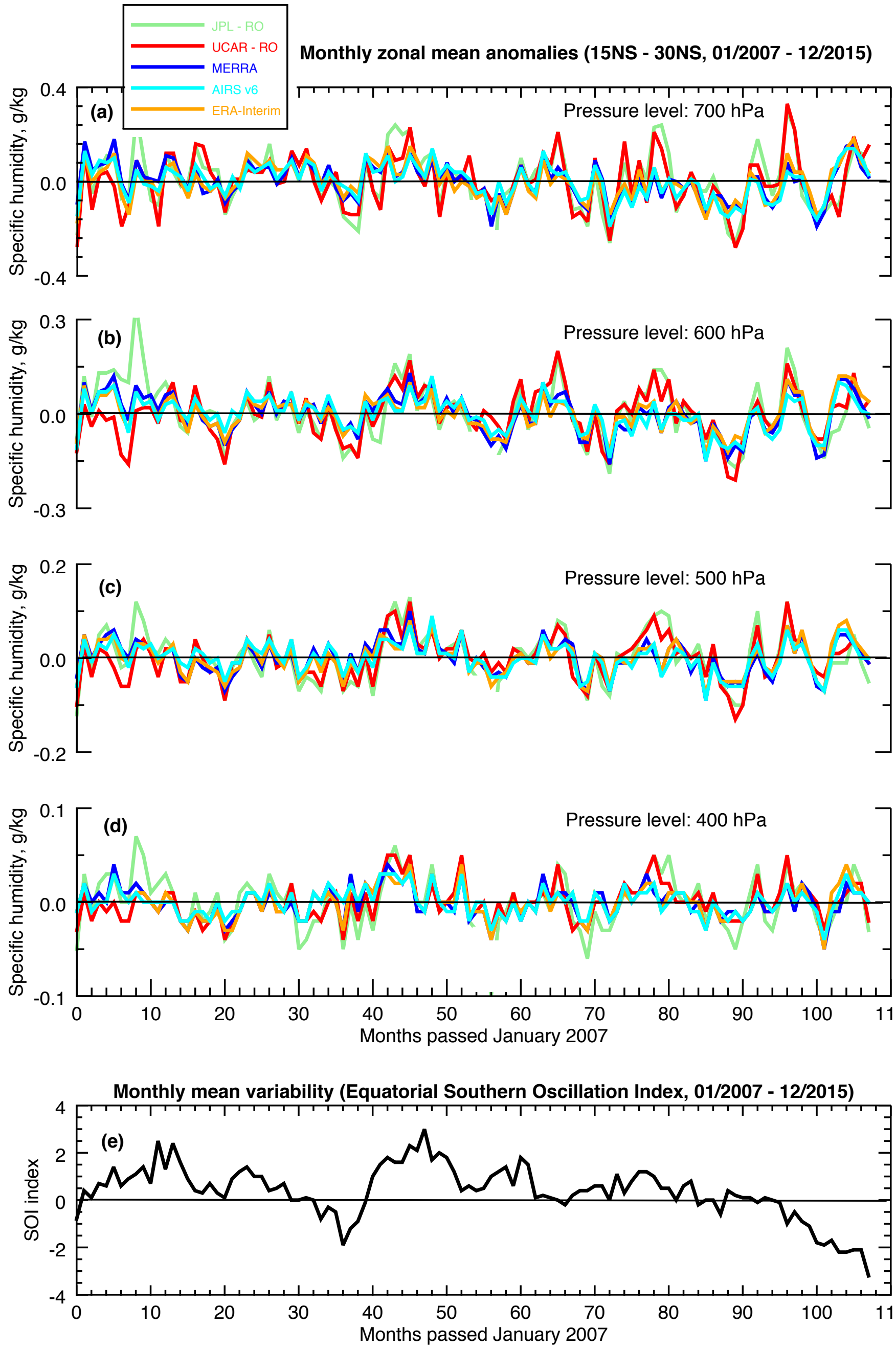

**Figure 5.** *Same as figure 3, but the results reflect SH trends in the 15NS–30NS latitudinal belt.*

The SH interannual anomalies of all data sets at 400 hPa are correlated (~ 0.6) with those at 700 hPa, but have smaller amplitude. The strength of their correlation over the trade winds zone is weaker and decreases with altitude compared to that estimated for the deep tropics. We suggest that this may be linked to the strength of the convection over the trade winds zone, which is weaker than that found over the deep tropics; thus, establishing a weaker vertical connection. Unlike the deep tropics, the SH interannual anomalies of all data sets show a weak cross-correlation (< 0.3) with the monthly mean SOI, when using a 5–month lag (and the cross-correlation is even smaller at 0–month lag).

Based on a linear regression fit and a Student *t*-test statistical analysis (criteria: $p < 0.05$ and 2-sigma) of the SH interannual anomalies, unlike the deep tropics, all data sets indicate no change in the amount of the SH up to 400 hPa with time (cf., Table 2). Contrary to the deep tropics, the linear regression fit slopes are negative, with the MERRA and AIRS data sets indicating a gradual SH decrease in the lower troposphere at 700 hPa and 600 hPa.

### 3.3. Analysis of the SH at the subtropics

The ±30-40° latitude belt, in both hemispheres, defines the subtropics where dry air masses descend from the Hadley cell. These moderate-to-strong subsidence regions exhibit low cloud formation (especially during the summer months), while favoring formation of low-altitude marine boundary layer (MBL) clouds.

In the subtropics, the interquartile range and 1-sigma uncertainty of the MERRA, ERA–Interim, and AIRS data sets at 700 hPa and 600 hPa is ~ 50% larger than those estimated for the deep tropics and the trade winds zones (cf., figure 1; bottom row), indicating much larger variability of the monthly zonal mean SH in the lower troposphere over dry air regions.

**Table 3.** *Same as Table 1, but for the subtropics ±30-40$^o$ region.*

| **PART I:** | **9–Year long mean of SH climatology with 2-sigma uncertainty, g/kg** | | | | |
|---|---|---|---|---|---|
| **Data Records** | **JPL–RO** | **UCAR–RO** | **ERA–Interim** | **MERRA** | **AIRS** |
| 400 hPa | 0.64 ± 0.12 | 0.44 ± 0.08 | 0.46 ± 0.10 | 0.42 ± 0.12 | 0.37 ± 0.08 |
| 500 hPa | 1.01 ± 0.26 | 0.88 ± 0.22 | 0.94 ± 0.28 | 0.92 ± 0.18 | 0.82 ± 0.26 |
| 600 hPa | 1.59 ± 0.36 | 1.44 ± 0.34 | 1.62 ± 0.52 | 1.61 ± 0.48 | 1.48 ± 0.50 |
| 700 hPa | 2.44 ± 0.52 | 2.25 ± 0.52 | 2.50 ± 0.64 | 2.64 ± 0.68 | 2.38 ± 0.76 |
| **PART II:** | **9–Year long mean of deviations from JPL–RO, g/kg** | | | | |
| 400 hPa | n/a | - 0.26 | - 0.24 | - 0.28 | - 0.32 |
| 500 hPa | n/a | - 0.13 | - 0.07 | - 0.09 | - 0.20 |
| 600 hPa | n/a | - 0.15 | + 0.03 | + 0.02 | - 0.11 |
| 700 hPa | n/a | - 0.19 | + 0.06 | + 0.20 | - 0.06 |
| **PART III:** | **Linear regression fits of SH anomalies with 2-sigma uncertainty, g/kg/month** | | | | |
| 400 hPa | (-1.3±2.0)x10$^{-4}$ | (1.1±1.0)x10$^{-4}$ | (1.1±0.8)x10$^{-4}$ | (1.0±0.8)x10$^{-4}$ | (0.8±0.8)x10$^{-4}$ |
| 500 hPa | (-1.4±2.4)x10$^{-4}$ | (1.1±2.0)x10$^{-4}$ | (1.6±1.6)x10$^{-4}$ | (0.3±1.6)x10$^{-4}$ | (0.4±1.4)x10$^{-4}$ |
| 600 hPa | (-2.0±4.2)x10$^{-4}$ | (2.8±3.4)x10$^{-4}$ | (2.1±2.6)x10$^{-4}$ | (0.4±2.8)x10$^{-4}$ | (-3.1±2.2)x10$^{-4}$ |
| 700 hPa | (-0.3±5.8)x10$^{-4}$ | (3.9±4.6)x10$^{-4}$ | (4.0±3.6)x10$^{-4}$ | (2.9±4.0)x10$^{-4}$ | (-4.5±3.2)x10$^{-4}$ |

The interquartile ranges for the JPL–RO and UCAR–RO data sets do not show any differences among the three climate zones, suggesting that RO observations show smaller variability in the SH than the two reanalyses and the AIRS data sets regardless of the climate zone and dynamics.

At the subtropics, similar to the trade winds zones, at 700 hPa, the JPL–RO data set agrees very well with the ERA–Interim and the AIRS data sets within < 2.5% (statistically insignificant), and but is drier than MERRA by 8.0% (statistically significant). Again, the UCAR–RO data set is the driest among all data sets by ~ 8.0% (statistically significant); however, during the autumn and winter seasons it agrees very well with the AIRS observations throughout the entire time period (cf., figure 6) but during spring and summer AIRS captures wetter air than UCAR–RO. Moving higher into the troposphere, at 600 hPa, the JPL–RO data set

agrees very well with both reanalyses differing by < 2.0% (statistically insignificant), but it is now statistically wetter than the AIRS data set by ~ 7.0%. In particular, the JPL–RO data sets capture almost the same month–to–month zonal mean SH values with the two reanalyses during autumn and winter, and the AIRS data set is in excellent agreement with the UCAR–RO data set during the same seasons. The UCAR–RO data set continues to be the driest among all data sets by > 10% with respect to both reanalyses and the JPL–RO data set, but it is statistically the same with AIRS differing by < 3.0%.

In the middle troposphere, the JPL–RO data set starts indicating that the air is moister than all other data sets by > 4.0%, and this wetness becomes much more pronounced at 400 hPa with the JPL–RO data set indicating that the atmosphere is wetter by > 30% with respect to the rest of the data sets. The JPL–RO time series defines the maximum SH values at 500 hPa and 400 hPa, while the AIRS data set sets the minimum SH values at the respective pressure levels, with the two reanalyses and the UCAR–RO data sets lay in between the JPL–RO and the AIRS data sets. MERRA and ERA–Interim are statistically in excellent agreement with one another at 500 hPa differing by ~ 2.0%. The UCAR–RO data set is systematically drier with respect to the two reanalyses during the summer season by ~ 7.0% (with ERA–Interim) and ~ 4.5% (MERRA). This dryness might be causing the UCAR–RO data set to appear statistically different with respect to MERRA and ERA–Interim. At 400 hPa, all data sets are statistically different from one another within 2-sigma; yet, the UCAR–RO data set is in close agreement with MERRA (during spring and summer) and ERA–Interim (during autumn and winter).

414

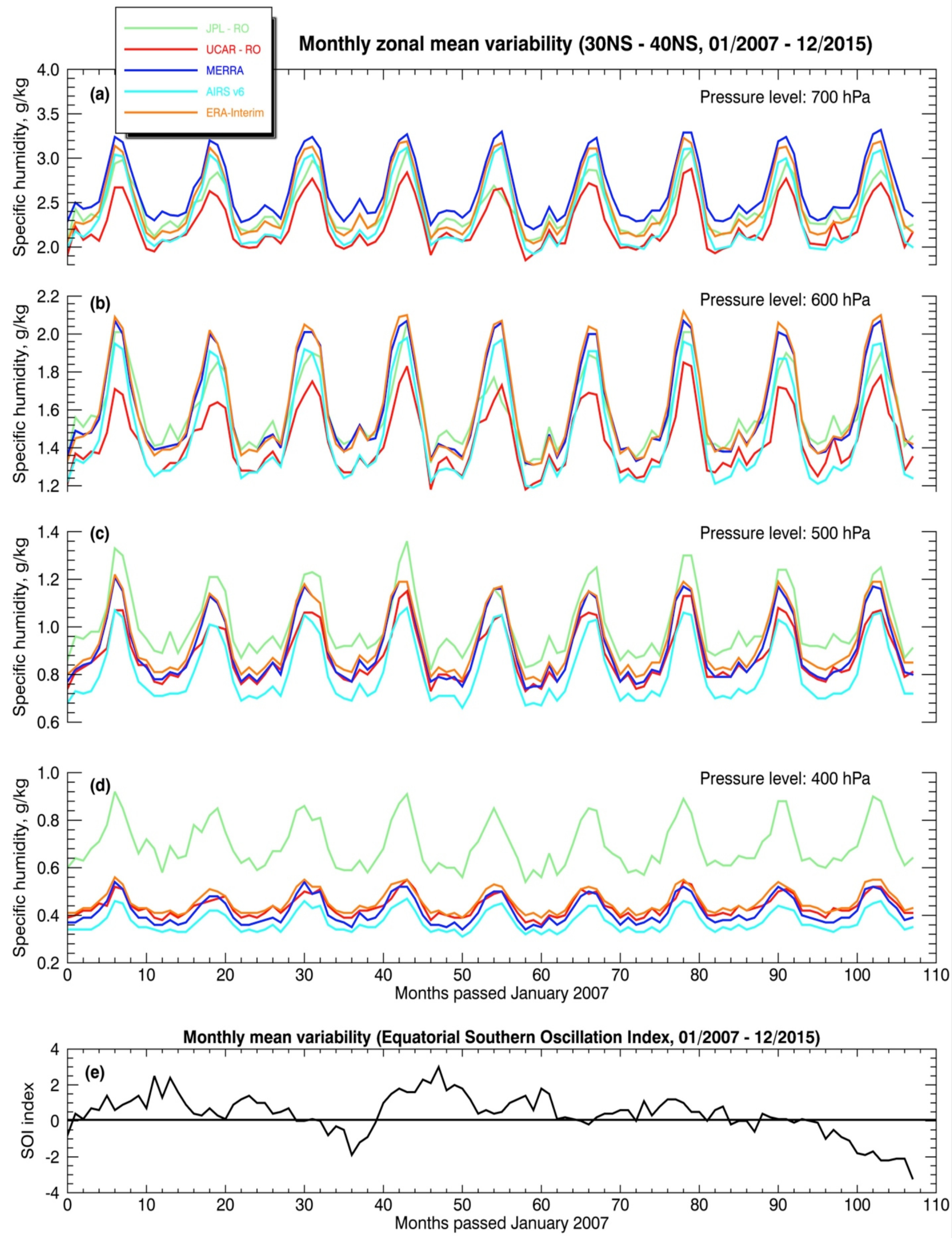

415 **Figure 6.** *Same as figure 2, but the results reflect SH trends in the subtropics at 30NS–30NS.*

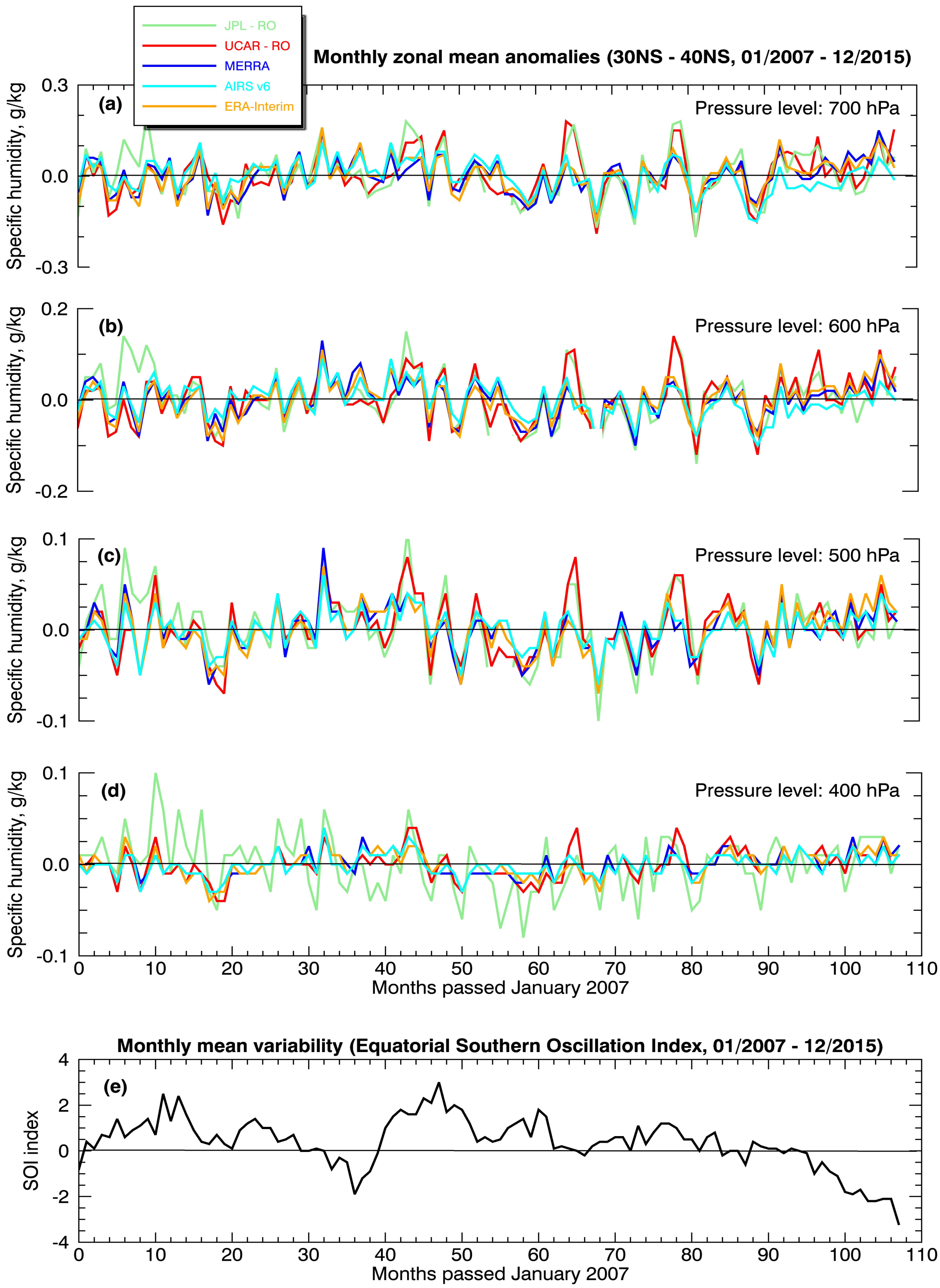

**Figure 7.** *Same as figure 3, but the results reflect SH trends in the subtropics ±30–40NS region.*

Considering that both MERRA and ERA–Interim under-predict the total cloud fraction over subsidence regions [*Dolinar et al.,* 2016], the two reanalyses might underestimate the air absolute humidity. Additionally, taking into account the fact that JPL RO retrieval technique uses ECMWF as *a-priori* temperature information in the forward refractivity operator to estimate the SH (that has a warm bias in the upper troposphere) implies that JPL–RO may be overestimating the SH at 400 hPa. The abovementioned arguments might explain (partly) the > 30% disagreement of the JPL–RO data set with respect to all other the data sets, and the current results show that the JPL–RO time series senses a wetter subsidence zone throughout the troposphere – more so at 400 hPa.

Compared to the deep tropics and the trade winds zones, the absolute differences of the SH values averaged over the entire time period between the JPL–RO and the rest of the data sets throughout the vertical extent of the troposphere are smaller than in the deep tropics and similar to the trade winds zone, except at the 400 hPa where it remains almost the same. Again, this hints towards the notion that different data sets agree better with one another over regions characterized by less convection. The monthly zonal mean SH variability also shows a clear annual cycle signature throughout the vertical extent of the troposphere, but the amplitudes of the SH interannual anomalies is ~ 30–50% smaller (cf., figure 5) than those estimated over the trade winds zone.

The SH interannual anomalies of all data sets at 400 hPa are again correlated (~ 0.65), with the anomalies at 700 hPa, with their amplitudes decreasing with altitude. The strength of their correlation over the subtropics is similar to that estimated over the trade winds zone and weaker than that found over the deep tropics. Again, this may hint that the strength of the convection is coupled with the correlation strength of the SH anomalies throughout the vertical

extent of the troposphere. Unlike the deep tropics, the SH interannual anomalies of all data sets show a weak cross-correlation ranges from ~ 0.25 (at 700 hPa and 600 hPa) to ~ 0.4 (at 500 hPa and 400 hPa) with the monthly mean SOI. This indicates that there is a connection between the surface temperature variability and the atmosphere aloft, and the surface climate variability affects more the upper than the lower troposphere. The magnitude of the correlation at the subtropics is smaller than that found in the deep tropics, suggesting that convection may be key to establishing the extent and strength of vertical teleconnection in the troposphere.

Based on a linear regression fit and a Student *t*-test statistical analysis (criteria: $p < 0.05$ and 2-sigma) of the SH interannual anomalies, unlike the trade winds zones, ERA–Interim and UCAR–RO (at all pressure levels) and AIRS (at 500 hPa and 400 hPa) show moistening of the subtropics, except from the AIRS 700 hPa and 600 hPa pressure levels where the data sets indicate a decrease in the SH over time. The JPL–RO data sets neither does it show decrease or increase of SH with time, and MERRA shows moistening of the upper troposphere.

## 4. Conclusions

We conclude that based on statistical tests using a 2-sigma uncertainty and 95% confidence level criteria the RO observations: (a) capture similar patterns of the monthly zonal mean SH annual variability and trend as the two reanalyses and the AIRS observations (except from the JPL-RO time series that exhibit discrepancies in the SH variability at the beginning of the year 2007 and in the summer of 2011). (b) They capture the same SH annual cycle signature as all other data sets. (c) The RO interannual anomalies are in excellent agreement with all other data sets, both in magnitude and variability, despite discrepancies in the absolute value of SH with respect to other data sets. (d) The SH differences between JPL and UCAR are variable

depending on location and pressure level, ranging in general between 5.0% and 15.0%. This difference, although it is statistically significant at the 95% confidence level, falls within JPL's retrieval uncertainty [*Vergados et al.,* 2014]. Given the above, the RO observations could augment the reanalyses and satellite observations by providing an independent data set to study short-term SH variations, which are critical to the study of water vapor trends, and climate sensitivity, variability, and change. Although RO observations capture very well the SH variabilities and trends with time, we ought to point out that there exist discrepancies among the data sets over certain seasons and climate regions that introduce statistically significant differences in the amount of tropospheric SH measured by each data set.

In the middle-to-upper troposphere, at 500 hPa and 400 hPa, we notice that over all climate zones (despite the convection strength), the JPL–RO data set is the moistest than all data sets, the AIRS data set is the driest than all data sets, and the UCAR–RO data set agrees very well with both the ERA–Interim and MERRA reanalyses. Given the AIRS dry bias in the upper troposphere [*Fetzer et al.,* 2008], potential warm temperature bias in the JPL retrieval algorithm, and the fact that the UCAR–RO variation assimilation uses ERA–Interim as *a-priori*, we could explain part of the observed differences and data set agreement. We must point out that the JPL–RO observations systematically show moister air during the summer throughout the entire time period, which could also explain the observed overall wet bias with respect to the rest of the data sets. Over the deep tropics, the UCAR–RO and ERA–Interim data sets show a positive trend in the SH interannual anomalies at the 95% confidence level, but the rest of the data sets indicate no trend. Over the trade winds zones, all data sets indicate no trend in the SH interannual anomalies at the 95% confidence level. Over regions of strong subsidence, the JPL–RO and AIRS data sets

do not indicate any trend in the SH interannual anomalies, but the UCAR–RO and the two reanalyses suggest a positive trend.

Unlike the middle-to-upper troposphere, where the agreement and disagreement among data sets is consistent over all climate zones, in the lower-to-middle troposphere there is a complex behavior of discrepancies. We speculate that this might be because the 700 hPa pressure level lies above the planetary boundary layer that interfaces with the free troposphere via convection and entrainment. This implies that the SH measured by each data set might be susceptible to the degree which each data set represents this vertical coupling.

In particular, over the ±15$^{o}$ (where the troposphere is subject to deep convection), the JPL–RO observations agree very well with MERRA (which does not assimilate ROs), while the UCAR–RO, ERA–Interim, and AIRS agree much better with one another. We argue here that ERA–Interim produces less total cloud fraction than MERRA. Considering that UCAR–RO and AIRS use ERA–Interim as *a-priori*, we might explain why UCAR–RO, ERA–Interim, and AIRS capture drier air than MERRA. Although the comparison between the JPL–RO and UCAR–RO data sets is not the focus of this study, considering the above discussion, we could argue that because the JPL–RO and MERRA data sets are independent measurements, the UCAR–RO, ERA–Interim, and AIRS underestimate the amount of SH during deep convection in the lower troposphere. Over the trade winds zones, in the ±15–30$^{o}$, the JPL–RO observations are in very good agreement with ERA–Interim, AIRS, and MERRA (except at 700 hPa), whereas the UCAR–RO observations are again drier than all data sets. Over the subtropics, where dry air masses descend through the Hadley cell, the JPL–RO observations agree very well with MERRA and ERA–Interim, while the UCAR–RO data set agrees better with AIRS.

In all climate zones the UCAR–RO, together with the AIRS data set, systematically show drier air in the lower troposphere than all other data sets. Aside from the AIRS low-cloud contamination [*Schreier et al.,* 2014], this behavior could indicate that both AIRS and UCAR–RO data sets may not be sensitive enough to properly capture high-moisture air rising from the boundary layer beneath, either due to entrainment and/or convective limitations. This study exploits the short-term RO SH data record in an attempt to quantify differences between the RO time series and other data sets. More detailed statistical analysis is required between the SH products among different RO processing centers to define its structural uncertainty. The reduced daily sampling of the COSMIC missions may be also a limiting factor in properly establishing differences between the RO and other platforms. We expect that the increased sampling rate of the COSMIC-2 follow-on mission will provide a much better picture of the tropical and subtropical SH climatology, which will help us extend the current short-term RO SH record.

**Acknowledgments:**

This research was carried out at the Jet Propulsion Laboratory, California Institute of Technology, under a contract with the National Aeronautics and Space Administration Earth Science Mission Directorate (SMD). We thank Robert Khachikyan for making publicly available the JPL-RO retrievals through the AGAPE interactive search tool. We would like to acknowledge the University Corporation for Atmospheric Research (UCAR) COSMIC Data Analysis and Archive Center (CDAAC) for making publicly available the COSMIC data sets. We would like to thank NASA Earth Observing System Data and Information System (EOSDIS) for making publicly available the MERRA and AIRS data sets. Finally, we would like to thank the European Center for Medium-range Weather Forecasts (ECMWF) for making publicly available the ERA–Interim data sets.

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
