# Peer review of "Manuscript under review for journal Atmos. Meas. Tech."

_Atmospheric Measurement Techniques, 2017_

## Referee Comment (RC1) · R. Anthes (Referee) · 13 Sep 2017

Overall this is an interesting and generally well-written paper that should be published in AMT. There have been relatively few papers published comparing water vapor retrievals from radio occultation (RO) observations to other independent observations and with each other (computed using different methods). Thus this is an important contribution.

However, the paper is long and it is a little difficult and tiresome to read because there are three regions (+/-15 degrees NS, 15-30 NS and 30-40 N/S and these are discussed in great detail with two figures and one table for each region. All of this takes about 16 pages and the reader may get lost. Perhaps the number of regions could be reduced to

two (0-30 NS and 30-40 N/S? It is not clear to me that the differences between +/- 15NS and 15-30 NS are important. I become lost in the details of all these comparisons.

Most importantly (and the reason for suggesting a major revision before publication), because a major point of the paper is a comparison of the JPL and UCAR retrievals of specific humidity q (and these are quite different) it is worth mentioning in the abstract the significant difference between the JPL and UCAR estimation of q given refractivity N. JPL uses a "simple" method (using T from the ECMWF TOGA database in Eq. 1) while UCAR uses a 1DVAR method (using ERA-Interim for the a priori). This difference between these two methods is likely the main reason for the different results, and not a property of RO in general. A reader could conclude from these large differences in specific humidity that RO is not a very good climate observational technique, unlike the results from many other studies. However, this reason should be verified by also comparing the JPL and UCAR refractivities that were used in computing the q. Or use the same method for computing q given N for UCAR as for JPL (i.e. use Eq. 1 for computing the UCAR estimate of q. This would narrow the differences to differences in refractivity.

Finally, it would be helpful if the authors could say something about what all these differences mean in terms of accuracy of water vapor compared to the estimates of accuracy in q from other papers. Perhaps this discussion could go in the conclusions.

Specific comments:

1. SH is not a common abbreviation for specific humidity. I suggest using the more common letter "q." 2. Line 32-something missing here? "as well as" perhaps? 3. Page 10 lines 206-215. The quoted RO accuracies of 10-20% below 7 km and 0.1 g/kg seem inconsistent. For a typical lower tropospheric q of 5-10 g/kg, an error of 0.1 g/kg (1-2%) is far better than 10% (1-2%). The JPL quoted accuracies of 0.2-0.4 g/kg in the tropics (2-4% for a typical value of q of 10 g/kg) are also very high compared to the quoted values of 20% for MERRA and 25% for AIRS. Can the authors comment on these large

differences? In general, it is very important for this paper to precisely define previous studies of the accuracy of water vapor (specific humidity) estimates from RO. 4. It would be helpful to know why the author's study extends downward only to 700 hPa? Most of the atmospheric water vapor is below 700 hPa. Yes, there is the negative N bias associated with super-refraction and other issues in the lower troposphere, but still it is important to characterize the errors in retrieved q in this region. 5. The Vergados et al. 2016 paper is in the list of References, but I couldn't find it mentioned in the paper. 6. Lines 285-287—It says that the wet bias in JPL-RO may be due to the warm bias in the ERA-Interim (Eq. 1). But they use ECMWF TOGA analysis for the T in Eq. 1, not the ERA-Interim (lines 150-151). Please clarify. Similarly, lines 420-422 say the JPL retrieval technique uses "ECMWF" as a-priori temperature information. What ECMWF, TOGA or Interim? 7. Figure 3 is not referred to in text. It looks like it should be in line 291, i.e. ". . .we estimate the respective SH anomalies (Figure 3)." 8. Lines 372-373-I suggest rewording to . . .defines the subtropics where dry air descends from the Hadley cell." 9. Lines 474-475-reword to say "moistest of all data sets" and "driest of all data sets." 10. Lines 490-492-All the pressure levels lie above the PBL not just the 700 hPa level. Do the authors mean that the 700 hPa level is the closest to the PBL?

End of comments

---

## Referee Comment (RC2) · Anonymous Referee #2 · 14 Sep 2017

General comments:

The paper compares specific humidity derived from GPS RO and different weather model reanalysis. Although the results might not be ground-breaking, they provide valuable information. I have some minor comments and questions.

Minor comments:

Abstract:

P2, L38: '...together with the retrieval uncertainty of the SH products from all data sets, we conclude that RO observations are a valuable independent observing sys-

tem.' What do you mean by 'independent'? RO SH is not independent from weather model data. JPL-RO SH makes use of the temperature from ECMWF. UCAR-RO SH is obtained by variational data assimilation utilizing ECMWF as the background. I suggest to remove the word 'independent'. Also, ECMWF depends on RO, because UCAR-RO bending angles were assimilated.

Introduction:

P3,L48: '...Hence, we ought to quantify and understand the degree of agreement of the water vapor concentration throughout the vertical extent of the troposphere among different sensors, in order to improve the representation of the Earth's atmospheric humidity content that is key to predicting future climate [Hegerl et al., 2015].' In the present study you consider the altiude range 700 -400 hPa ($\sim$2 – 8 km). The troposphere extents from $\sim$0 – 15 km. In fact, most of the water vapor is contained in the lowest 2 km. In the present study you do not try to quantify and understand the degree of agreement of the water vapor concentration throughout the vertical extent of the troposphere. I suggest to remove the word 'throughout'.

P4,L38: '...and full diurnal cycle sampling.' This is approximately true for COSMIC but not true in general. This depends on the LEO orbits.

P5,L102: '...Of importance is the fact that we use MERRA, instead of MERRA-2, because MERRA does not assimilate ROs (unlike ERA–Interim), providing an independent data set when comparing the RO SH observations.' This sounds interesting. Does this mean that you expect big differences when you use MERRA-2 instead of MERRA? Would it be a lot of effort for you to add MERRA-2 as well? I recommend to do so. This would be very interesting, because it would show the impact of RO on weather model SH.

Methodology:

P6, L114: '...We study the tropics and subtropics ($\pm$40o, in three distinct latitudinal

regions) from 700 hPa up to 400 hPa, because this region is key to climate research [IPCC, 2007], but models and observations have large SH differences in the middle and upper troposphere [e.g., Jiang et al., 2012; Tian et al., 2013; Wang and Su, 2013], and we select this pressure range because the RO SH retrievals are most robust.' I can imagine what you mean by 'most robust' but some other interested readers do not know what this means. Please, explain in brief what you mean by 'most robust'. E.g. signal tracking in the lower troposphere is somewhat problematic, the assumption of a spherically layered atmosphere, critical refraction (Ao et al., 2003) etc.

P7, L144: '...air temperature'. I suggest to remove the word 'air'.

P7, L145: Please add (for completness) the equation that you use to convert water vapor pressure to SH.

P7, L154: '...air refractivity'. I suggest to remove the word 'air' here and in the following.

P9, L188: '...The AIRS physical retrievals use an IR–microwave neural net solution [Blackwell et al., 2008] as the first guess for temperature and water vapor profiles based on MIT's stochastic cloud-clearing and neural network solution described in Khan et al. [2014].' I have very little idea of AIRS retrieval. In short, does the AIRS retrieval at any point make use of data from a climatology or a weather model?

P9, L192: The section 'Data Sources' can be moved to the Acknowledgments.

P10, L207: '...GPS-RO air refractivity accuracy of <1.0% at 2.0 km altitude [Schreiner et al., 2007] reduces to ∼0.2% above 5.0 km [Kuo et al., 2005].' Schreiner et al., 2007 provides an estimate for the precision and not the accuracy. They measure the degree of the reproducibility of the GPS RO technique. Kuo et al., 2005 provide an estimate for the accuracy. As you focus on the altitude range 2 – 8 km, I suggest to simply write: 'GPS-RO refractivity accuracy is about 1% at an altitude of 2 km and decreasing to about 0.2% at an altitude of 8 km [Kuo et al., 2005].'

Results and discussion:

P10, L223: I suggest to remove '...We do not extend our analysis at higher altitudes due to the small contribution of water vapor on to the RO observations.' as you already mention in the 'Methodology' section that your focus is 700-400 hPa.

P11, L226: '...and the differences between the JPL and the UCAR time series serve as a guideline of an estimate of the SH structural uncertainty.' One of the most interesting points in your study are the differences between JPL SH and UCAR SH. Where do the differences come from? Are those differences due to differences in the raw (=non-optimized) bending angles, the refractivity or are they mainly caused by the different SH retrieval method? I strongly recommend to add (in an Appendix) a one-to-one comparison (mean and one-sigma) for bending angle and refractivtiy profiles for the altitude range 0-8 km.

P12, L240: '...SH time series over the entire observational record for all data sets throughout the vertical extent of the troposphere'. Remove the word 'throughout'.

P18, L332: '...Overall, this suggests that over less convective regions different data sets tend to agree better, signifying that convection is a limiting factor in properly sensing the amount of water vapor in the atmosphere.' Weather models are known to be less accurate in regions with convection. Do you mean that RO SH is less accurate there aswell? For example there is one study by S. Yang and Zou, 2017 showing (positive) RO biases in cloudy conditions.

S. Yang and Zou (2017) Dependence of positive refractivity bias of GPS RO cloudy profiles on cloud fraction along GPS RO limb tracks, GPS Solut, 21:499–509 DOI 10.1007/s10291-016-0541-1

P26, L421: Remove 'in the forward operator'.

Conclusion:

P28, L467: I suggest to remove the word 'independent'. RO (non-optimized) bending angles are independent, however RO SH is not independent.

---

## Author Comment (AC1) · 7 Dec 2017

**Manuscript Number** : amt–2017–250-RC2
**Associate Editor** : Dr. Jens Wickert
**Manuscript Title** : Comparisons of the tropospheric specific humidity from GPS radio occultations with ERA-Interim, NASA MERRA and AIRS data

**Dear Referee #1,**

We would like to thank reviewer #1 for taking the time to review our manuscript. We greatly appreciate all comments, which we address and implement in the revised manuscript. The manuscript has now become stronger and presents additional results for discussion reflecting the reviewer's comments.

**General Comment #1:** The paper is long and it is a little difficult and tiresome to read because there are three regions and these are discussed in great detail with two figures and one table for each region. All of this takes 16 pages and the reader may get lost. Perhaps the number of regions could be reduced to two? It is not clear to me that the difference between +/- 15NS and 15-30NS are important. I become lost in the details of all these comparisons.

**Answer:** Agreed. However the 500 hPa and 400 hPa show the same behavior in all three regions. The only difference is found at the 700 hPa and 600 hPa, which are most influenced by convection. Thus, although we agree that analyzing three different regions is tiresome, we want to be inclusive and decided not to merge the results from the +/- 15NS and 15-30NS regions into one. This is because we would have missed seeing the different behavior of the data at 700 hPa and 600 hPa in the two regions. However, we took the following actions to make the results easier to read:

**Actions taken:**
1.  We only show the monthly zonal mean time series of the specific humidity and their interannual anomalies and the accompanied table for the deep tropics (+/- 15NS) and moved the rest of the figures and tables into the supplementary material. However, we kept their discussion in the text.

2.  We written more concisely the analysis for each region and avoided repetitive discussion at 500 hPa and 400 hPa pressure levels, focusing only in the lower troposphere.
* * *
**General Comment #2:** Most importantly, because a major point of the paper is a comparison of the JPL and UCAR retrievals of specific humidity it is worth mentioning in the abstract the significant difference between the JPL and UCAR estimation of q given refractivity N. JPL uses a "simple" method (using T from ECMWF TOGA database in Eq. 1) while UCAR uses a 1DVAR method (using ERA-Interim for the a priori). This difference between these two methods is likely the main reason for the different results, and not a property of RO in general. This reason should be verified by also comparing the JPL and UCAR refractivities that were used in computing q.

**Answer:** The reviewer is correct.

**Actions taken:**
1.  We added relevant text in the manuscript to explicitly state this. **See Abstract lines 31–33, and lines 148–153.**

2.  We performed additional data processing and data analysis for the refractivity climatologies and included the results in the manuscripts in a new section and discussion. **See new Section 3.4.**

**General Comment #3:** Finally, it would be helpful if the authors could say something about what all these differences mean in terms of accuracy of water vapor compared to the estimates of accuracy in q from other papers. Perhaps this discussion could go in the conclusions.

**Answer: Done.** We included background information about the accuracy of RO q retrievals and compare them with the accuracy of other data sets. Based on this discussion, we explicitly discuss about the statistical significance of our results throughout the manuscript (when comparing the different climatologies). **See new added Section 3.4 and lines 235–236.**
* * *
**Specific Comment #1:** SH is not a common abbreviation for specific humidity. I suggest using the more common letter "q".

**Answer:** Agreed. We removed the abbreviation SH from the manuscript. Instead, we explicitly write "specific humidity".
* * *
**Specific Comment #2:** Line 32. Something is missing here? "as well as" perhaps?

**Answer: Done. Sentence was modified. No need to act on this any more.**
* * *
**Specific Comment #3:** Page 10, lines 206 – 215. The quoted accuracies of 10-20% below 7 km and 0.1 g/kg seem inconsistent. For a typical lower tropospheric q of 5-10 g/kg, an error of 0.1 g/kg (1-2%) is far better than 10% (1-2%). The JPL quoted accuracies of 0.2-0.4 g/kg in the tropics (2-4% for a typical value of q of 10 g/kg) are also very high compared to the quoted values of 20% for MERRA and 25% for AIRS. Can the authors comment on these large differences? In general, it is very important for this paper to precisely define previous studies of the accuracy of water vapor (specific humidity) estimates from RO.

**Answer: Done.** We devoted a separate section establishing the RO specific humidity accuracies based on previous studies. **See Section 2.6**
* * *
**Specific Comment #4:** It would be helpful to know why the author's study extends downward only to 700 hPa? Most of the atmospheric water vapor is below 700 hPa. Yes, there is negative N bias associated with super-refraction and other issues in the lower troposphere, but still it is important to characterize the errors in retrieved q in this region.

**Answer:** This is the same comment with that of Reviewer #2 Minor Comment #5. The reason is exactly what the reviewer mentions above. Also, the spherical symmetry approximation and signal tracking issues could also play a role here. In this preliminary climatology analysis, we wanted to focus on the pressure range that we are confident the RO humidity is well established, and then we would focus on the boundary layer and higher up in the troposphere. **We have added relevant text to clarify this.** See lines 121–127 and Conclusion section.**
* * *
**Specific Comment #5:** The Vergados et al. 2016 paper is in the list of references, but I could not find it mentioned in the paper.

**Answer: Done.** We removed the references.
* * *
**Specific Comment #6:** Lines 285-287. It says that the wet bias in JPL-RO may be due to the warm bias in the ERA-Interim (We. 1). But they use ECMWF TOGA analysis for the T in Eq. 1, not the ERA-Interim (lines 150-151). Please clarify. Similarly, lines 420-422 say the JPL retrieval technique uses "ECMWF" as a-priori temperature information. What ECMWF, TOGA or Interim?

**Answer: Done. See line 165 and lines 495–500.**
* * *
**Specific Comment #7:** Figure 3 is not referred to in text. It looks like it should be in line 291, i.e. "...we estimate the respective SH anomalies (Figure 3)."

**Answer: Done.** Due to re-arranging the figures, Figure 3 now shows the specific humidity anomalies at the deep tropics and is discussed throughout the manuscript.
* * *
**Specific Comment #8:** Lines 372-373. I suggest rewording to "...defines the subtropics where dry air descends from the Hadley cell."

**Answer: Done. See lines 423–424.**
* * *
**Specific Comment #9:** Lines 474-475. Reword to say "moistest of all data sets" and "driest of all datasets".

**Answer: Done. See lines 519–520.**
* * *
**Specific Comment #10:** Lines 490-492: All the pressure levels lie above the PBL not just the 700 hPa level. Do the authors mean that the 700 hPa level is the closest to the PBL?

**Answer: Yes. Please, see modified lines 522–523.**
* * *
*Panagiotis Vergados*

**THIS IS THE END OF REVIEWER #1 REPORT …………………………………………………..**

---

## Author Comment (AC2) · 7 Dec 2017

| Manuscript Number | : amt-2017-250-RC2                                                                   |
|-------------------|--------------------------------------------------------------------------------------|
| Associate Editor  | : Dr. Jens Wickert                                                                   |
| Manuscript Title  | : Comparisons of the tropospheric specific humidity from GPS radio occultations with |
|                   | ERA-Interim, NASA MERRA and AIRS data                                                |

**Dear Referee #2,**

We would like to thank you for taking the time to review our manuscript. Your kind words about our work are greatly appreciated, and your comments have now been addressed and implemented in the revised manuscript. We have performed major revisions to accommodate your Comment #13, and we include the results in the revised version.

**Minor Comment #1:** P2, L38: '... together with the retrieval uncertainty of the SH products from all data sets, we conclude that RO observations are a valuable independent observing system.' What do you mean by 'independent'? RO SH is not independent from weather model data. JPL-RO SH makes use of the temperature from ECMWF. UCAR-RO SH is obtained by variational data assimilation utilizing ECMWF as the background. I suggest to remove the word 'independent'. Also, ECMWF depends on RO, because UCAR-RO bending angles were assimilated.

\_\_\_\_\_

Answer: Done. We removed the word "independent".

**Minor Comment #2:** P3, L48: '...Hence, we ought to quantify and understand the degree of agreement of water vapor concentration throughout the vertical extent of the troposphere among different sensors, in order to improve the representation of the Earth's atmospheric humidity content that is key to predicting future climate [Hegerl et all., 2015].' In the present study you consider the altitude range 700-400 hPa ( $\sim 2-8$  km). The troposphere extends from  $\sim 0-15$  km. In fact, most of the water vapor is contained in the lowest 2 km. In the present study you do not try to quantify and understand the degree of agreement of the water vapor concentration throughout the vertical extent of the troposphere. I suggest to remove the word 'throughout'.

Answer: Done. We removed the word "throughout". Please, see strikethrough in line 49.

**Minor Comment #3:** P4, L83: '...and full diurnal cycle sampling." This is approximately true for COSMIC but not true in general. This depends on the LEO orbits.

\_\_\_\_\_

Answer: Done. We added the reviewer's comment in the revised manuscript. Please, see lines 82–83.

**Minor Comment #4:** P5, L102: '...Of importance is the fact that we use MERRA, instead of MERRA-2, because MERRA does not assimilate (unlike ERA-Interim), providing an independent data set when comparing the RO SH observations.' This sounds interesting. Does this mean that you expect big differences when you use NERRA-2 instead of MERRA? Would is be a lot of effort for you to add MERRA-2 as well? I recommend to do so. This would be very interesting, because it would show the impact of RO on weather model SH.

Answer: We believe that adding the MERRA-2 SH climatology in our analysis will not show the impact of RO on weather model SH. This is because there have been significant changes on how MERRA-2 handles the Earth's water cycle with respect to MERRA, and these changes have a much more direct contribution to differences in MERRA-2 SH climatology than the addition of RO bending angles. Specifically, *Bosilovich et al.* [2017] state: "Some of the changes in MERRA-2 have direct effect on the water cycle." For detailed explanation of these changes please refer to *Galero et al.* [2016] and *Takacs et al.* [2016]. Thus, we believe that comparisons with MERRA are more informative than comparisons with MERRA-2 for the objectives of our investigations, unless the contributions of all improvements in MERRA-2 are first isolated from the contributions of RO. However, we acknowledge the fact that comparing MERRA-2 and RO could be an interesting task. We added relevant text to discuss this. Please, see lines 175–180.

**Minor Comment #5:** P6, L114: '...We study the tropics and subtropics ( $\pm 40^{\circ}$ , three distinct latitudinal regions) from 700 hPa up to 400 hPa, because this region is key to climate research [IPCC, 2007], but models and observations have large SH differences in the middle and upper troposphere [e.g., Jiang et al., 2012; Tian et al., 2013; Wang and Su, 2013], and we select this pressure range because the RO SH retrievals are most robust.' I can imagine what you mean by 'most robust' but some other interested readers do not know what this means. Please, explain in brief what you mean by 'most robust'. E.g. signal tracking in the lower troposphere is somewhat problematic, the assumption of a spherically layered atmosphere, critical refraction (Ao et al., 2003) etc.

Answer: We included relevant text and removed "most robust" to avoid confusion. Please, see lines 121–127.

Minor Comment #6: P7, L144: '...air temperature'. I suggest to remove the word 'air'.

Answer: Done. Please, see strikethrough word in line 153.

**Minor Comment #7:** P7, L145: Please add (for completeness) the equation that you use to convert water vapor pressure to SH.

\_\_\_\_\_

Answer: Done. Please, see lines 158–163.

Minor Comment #8: P7, L154: '...air refractivity'. I suggest to remove the word "air" here and in the following.

\_\_\_\_\_

Answer: Done. Please, see strikethrough line 168.

**Minor Comment #9:** P9, L188: '...The AIRS physical retrievals use an IR-microwave neural net solution [Blackwell et al., 2008] as the first guess for temperature and water vapor profiles based on MIT's stochastic cloud-clearing and neural network solution described in Khan et al. [2014].' I have very little idea of AIRS retrievals. In short, does the AIRS retrieval at any point make use of data from a climatology or a weather model?

Answer: The short answer is no. The first guess comes from a neural network, which is trained on 60 days of ECMWF during the first year or two of AIRS operations [personal communication with Eric Fetzer]. It does not retrieve water profiles whenever cloud fraction exceeds the 80%, and recently they developed a cloud-clearing algorithm which compares the irradiance of neighboring pixels to infer the water vapor content during clouds.

Minor Comment #10: P9, L192: The section 'Data Sources' can be moved to the Acknowledgments.

Answer: Done. Please, see Acknowledgments.

**Minor Comment #11:** P10, L207: ' ...GPS-RO air refractivity accuracy of <1.0% at 2.0 km altitude [Schreiner et al., 2007] reduces to ~0.2% above 5.0 km [Kuo et al., 2005].' Schreiner et al., 2007 provides an estimate for the precision and not the accuracy. They measure the degree of the reproducibility of the GPS RO technique. Kuo et al., 2005 provide an estimate for the accuracy. As you focus on the altitude range 2 - 8 km, I suggest to simply write: 'GPS-RO refractivity accuracy is about 1% at an altitude of 2 km and decreasing to about 0.2% at an altitude of 8 km [Kuo et al., 2005].'

Answer: Done. Please, see lines 230–231.

**Minor Comment #12:** P10, L223: I suggest to remove '...We do not extend our analysis at higher altitudes due to small contribution of water vapor on to the RO observations.' As you already mention in the 'Methodology' section that your focus is 700-400 hPa.

\_\_\_\_\_

\_\_\_\_\_

Answer: Done. The sentence has been removed.

**Minor Comment #13:** P11, L226: '...and the differences between the JPL and the UCAR time series serve as a guidline of an estimate of the SH structural uncertainty.' One of the most interesting points in your study are the differences between JPL SH and UCAR SH. Where do the differences come from? Are those differences due to differences in the raw (=non-optimized) bending angles, the refractivity, or they mainly caused by the difference SH retrieval method? I strongly recommend to add (in an Appendix) a one-to-one comparison (mean and one-sigma) for bending angle and refractivity profiles for the altitude range 0-8 km.

**Answer: Done. This is similar to General Comment #3 of Reviewer #1. See new added Section 3.4.**

The differences in the specific humidity retrievals result from a combination of different things. We have analyzed the refractivity climatologies from both JPL and UCAR at 700 hPa, 600 hPa, 500 hPa, and 400 hPa pressure levels, and have included these results in the main manuscript. We also translate the refractivity differences into specific humidity differences and discuss the discrepancies between JPL and UCAR within these differences. We show these results for the deep tropics. The analysis is exactly the same for the trade winds zones and the subtropics and therefore we have not repeated it.

\_\_\_\_\_

**Minor Comment #14:** P12, L240: '...SH time series over the entire observational record for all data sets throughout the vertical extent of the troposphere'. Remove the word 'throughout'.

Answer: Done. Please, see strikethrough in line 343.

**Minor Comment #15:** P18, L332: '...Overall, this suggests that over less convective regions different data sets tend to agree better, signifying that convection is a limiting factor in properly sensing the amount of water vapor in the atmosphere.' Weather models are known to be less accurate in regions with convection. Do you mean that RO SH is less accurate there as well? For example there is one study by S. Yang and Zou, 2017 showing (positive) RO biases in cloudy conditions.

\_\_\_\_\_

Answer: Done. Please, see lines 526–528.

Minor Comment #16: P26, L421: Remove 'in the forward operator'.

**Answer:** Done. Also removed in other places throughout the manuscript.

**Comment #17:** P28, L467: I suggest to remove the word 'independent'. RO (non-optimized) bending angles

\_\_\_\_\_

Answer: Done. We replaced the word 'independent' with the word 'additional'. Please, see line 530.

Panagiotis Vergados

\_\_\_\_\_

THIS IS THE END OF REVIEWER #2 REPORT .....

are independent, however RO SH is not independent.

---

## Author Comment (AC3) · 7 Dec 2017

Please, see the attached "Track Changes Enabled" document, which also demonstrates how we modified the manuscript to accommodate all the comments.

Please also note the supplement to this comment:
https://www.atmos-meas-tech-discuss.net/amt-2017-250/amt-2017-250-AC3-supplement.pdf

---

## Author Comment (AC5) · 7 Dec 2017

We have moved all figures and tables for the discussion of the +/-15-30NS and +/-30-40NS from the main manuscript to supplementary material.

Please also note the supplement to this comment: https://www.atmos-meas-tech-discuss.net/amt-2017-250/amt-2017-250-AC5-supplement.pdf
* * *

---

## Author Comment (AC6) · 7 Dec 2017

**Comparisons of the tropospheric specific humidity from GPS radio occultations with**

**ERA–Interim, NASA MERRA and AIRS data**

Panagiotis Vergados[1], Anthony J. Mannucci[1], Chi O. Ao[1], Olga Verkhoglyadova[1], and Byron

Iijima[1]

[1] Jet Propulsion Laboratory, California Institute of Technology, Pasadena, California, USA

**Corresponding author:** P. Vergados, Jet Propulsion Laboratory, M/S 138-310B, 4800 Oak

Grove Dr., Pasadena, CA, 91109, USA. (Panagiotis.Vergados@jpl.nasa.gov)

**Table S1.** This is the same as Table 1, but for the ±15-30$^o$ climate zone.

| PART I:   9–year long mean of specific humidity climatology with 2-sigma uncertainty, g kg$^{-1}$ | | | | | |
|---|---|---|---|---|---|
| **Data Records** | **JPL** | **UCAR** | **ERA–Interim** | **MERRA** | **AIRS** |
| 400 hPa | 0.64 ± 0.12 | 0.55 ± 0.08 | 0.57 ± 0.06 | 0.54 ± 0.10 | 0.49 ± 0.08 |
| 500 hPa | 1.22 ± 0.28 | 1.12 ± 0.24 | 1.17 ± 0.22 | 1.15 ± 0.24 | 1.07 ± 0.22 |
| 600 hPa | 2.17 ± 0.44 | 1.93 ± 0.38 | 2.13 ± 0.38 | 2.24 ± 0.42 | 2.09 ± 0.38 |
| 700 hPa | 3.44 ± 0.50 | 3.28 ± 0.54 | 3.48 ± 0.44 | 3.77 ± 0.44 | 3.48 ± 0.44 |
| **PART II:   9–year long mean of deviations from JPL, g kg$^{-1}$** | | | | | |
| 400 hPa | n/a | - 0.09 | - 0.07 | - 0.10 | - 0.16 |
| 500 hPa | n/a | - 0.11 | - 0.05 | - 0.07 | - 0.15 |
| 600 hPa | n/a | - 0.23 | - 0.02 | - 0.09 | - 0.07 |
| 700 hPa | n/a | - 0.16 | + 0.04 | + 0.33 | + 0.04 |
| **PART III:   Linear regression of specific humidity anomalies with 2-sigma uncertainty, g kg$^{-1}$ month$^{-1}$** | | | | | |
| 400 hPa | (-0.7±1.8)x10$^{-4}$ | (1.1±1.2)x10$^{-4}$ | (0.3±1.0)x10$^{-4}$ | (-0.3±1.0)x10$^{-4}$ | (-0.3±1.0)x10$^{-4}$ |
| 500 hPa | (-0.5±3.6)x10$^{-4}$ | (1.6±2.8)x10$^{-4}$ | (-0.1±2.2)x10$^{-4}$ | (-1.3±2.2)x10$^{-4}$ | (-1.9±2.0)x10$^{-4}$ |
| 600 hPa | (-6.9±6.6)x10$^{-4}$ | (1.8±4.8)x10$^{-4}$ | (-1.9±3.4)x10$^{-4}$ | (-5.0±3.8)x10$^{-4}$ | (-5.2±3.2)x10$^{-4}$ |
| 700 hPa | (-3.9±8.6)x10$^{-4}$ | (-0.4±7.2)x10$^{-4}$ | (-3.8±4.8)x10$^{-4}$ | (-7.5±4.6)x10$^{-4}$ | (-6.2±4.4)x10$^{-4}$ |

**Figure S1.** This is the same as figure 1, but for the 15NS–30NS latitudinal belt.

**Figure S2.** This is the same as figure 2, but for the 15NS–30NS latitudinal belt.

**Table S2.** This is the same as Table 1, but for the subtropics ±30-40° region.

| PART I:   9–Year long mean of specific humidity climatology with 2-sigma uncertainty, g kg$^{-1}$ | | | | | |
|---|---|---|---|---|---|
| **Data Records** | **JPL** | **UCAR** | **ERA–Interim** | **MERRA** | **AIRS** |
| 400 hPa | 0.64 ± 0.12 | 0.44 ± 0.08 | 0.46 ± 0.10 | 0.42 ± 0.12 | 0.37 ± 0.08 |
| 500 hPa | 1.01 ± 0.26 | 0.88 ± 0.22 | 0.94 ± 0.28 | 0.92 ± 0.18 | 0.82 ± 0.26 |
| 600 hPa | 1.59 ± 0.36 | 1.44 ± 0.34 | 1.62 ± 0.52 | 1.61 ± 0.48 | 1.48 ± 0.50 |
| 700 hPa | 2.44 ± 0.52 | 2.25 ± 0.52 | 2.50 ± 0.64 | 2.64 ± 0.68 | 2.38 ± 0.76 |
| **PART II:   9–Year long mean of deviations from JPL, g kg$^{-1}$** | | | | | |
| 400 hPa | n/a | - 0.26 | - 0.24 | - 0.28 | - 0.32 |
| 500 hPa | n/a | - 0.13 | - 0.07 | - 0.09 | - 0.20 |
| 600 hPa | n/a | - 0.15 | + 0.03 | + 0.02 | - 0.11 |
| 700 hPa | n/a | - 0.19 | + 0.06 | + 0.20 | - 0.06 |
| **PART III:   Linear regression of specific humidity anomalies with 2-sigma uncertainty, g kg$^{-1}$ month$^{-1}$** | | | | | |
| 400 hPa | $(-1.3\pm2.0)\times10^{-4}$ | $(1.1\pm1.0)\times10^{-4}$ | $(1.1\pm0.8)\times10^{-4}$ | $(1.0\pm0.8)\times10^{-4}$ | $(0.8\pm0.8)\times10^{-4}$ |
| 500 hPa | $(-1.4\pm2.4)\times10^{-4}$ | $(1.1\pm2.0)\times10^{-4}$ | $(1.6\pm1.6)\times10^{-4}$ | $(0.3\pm1.6)\times10^{-4}$ | $(0.4\pm1.4)\times10^{-4}$ |
| 600 hPa | $(-2.0\pm4.2)\times10^{-4}$ | $(2.8\pm3.4)\times10^{-4}$ | $(2.1\pm2.6)\times10^{-4}$ | $(0.4\pm2.8)\times10^{-4}$ | $(-3.1\pm2.2)\times10^{-4}$ |
| 700 hPa | $(-0.3\pm5.8)\times10^{-4}$ | $(3.9\pm4.6)\times10^{-4}$ | $(4.0\pm3.6)\times10^{-4}$ | $(2.9\pm4.0)\times10^{-4}$ | $(-4.5\pm3.2)\times10^{-4}$ |

**Figure S3.** This is the same as figure 1, but for the subtropics at 30NS–30NS.

**Figure S4.** This is the same as figure 2, but for the subtropics ±30–40NS region.

---

## Author Response (AR2)

**Manuscript Number**   **:** amt–2017–250
**Associate Editor**    **:** Dr. Jens Wickert
**Manuscript Title**     **:** Comparisons of the tropospheric specific humidity from GPS radio occultations with ERA-Interim, NASA MERRA and AIRS data

**Dear Editor,**

We would like to thank you for the opportunity to publish our manuscript in the Special Issue of the International Radio Occultation Working Group (IROWG) at the Atmospheric Measurement Techniques journal.

**General Comment:** We have now included the missing reference of Schmidt et al. [2005] in this uploaded version of the final manuscript.

*Panagiotis Vergados*

**THIS IS THE END OF REVIEWER #1 REPORT** ……………………………………………………………………..